# dsRNA formation leads to preferential nuclear export and gene expression

Ivo Coban[1], Jan-Philipp Lamping[1], Anna Greta Hirsch[1], Sarah Wasilewski[1], Orr Shomroni[2], Oliver Giesbrecht[1], Gabriela Salinas[2] & Heike Krebber[1✉]

When mRNAs have been transcribed and processed in the nucleus, they are exported to the cytoplasm for translation. This export is mediated by the export receptor heterodimer Mex67–Mtr2 in the yeast *Saccharomyces cerevisiae* (TAP–p15 in humans)[1,2]. Interestingly, many long non-coding RNAs (lncRNAs) also leave the nucleus but it is currently unclear why they move to the cytoplasm[3]. Here we show that antisense RNAs (asRNAs) accelerate mRNA export by annealing with their sense counterparts through the helicase Dbp2. These double-stranded RNAs (dsRNAs) dominate export compared with single-stranded RNAs (ssRNAs) because they have a higher capacity and affinity for the export receptor Mex67. In this way, asRNAs boost gene expression, which is beneficial for cells. This is particularly important when the expression program changes. Consequently, the degradation of dsRNA, or the prevention of its formation, is toxic for cells. This mechanism illuminates the general cellular occurrence of asRNAs and explains their nuclear export.

The discovery of pervasive transcription and the revelation of the large number of non-coding RNAs (ncRNAs) raised the question of their functionality[4,5]. ncRNAs are divided into small (shorter than 200 nucleotides) and long ncRNAs. The lncRNAs are represented globally in life and are in their entirety perhaps the least-well-understood class of transcripts. Importantly, the deregulation of lncRNAs has been associated with human diseases such as cancer and neurodegenerative diseases[6]. They are more like mRNAs than they are different: they are transcribed by RNA polymerase II with similar chromatin states and undergo 5′-capping, splicing and 3′-polyadenylation[7], but most lack an open reading frame and therefore also the coding potential of mRNAs[8]. Several reports indicate nuclear functions for lncRNAs in *S. cerevisiae* in regulating coding genes by suppressing transcriptional leakage[9–11] or by inducing transcription in response to environmental changes[12,13]. However, many antisense lncRNAs travel into the cytoplasm for reasons unknown, as becomes evident after mutation of the cytoplasmic exonuclease Xrn1 (ref. 3). Xrn1-mediated lncRNA degradation follows RNA translation and subsequent recognition through the non-sense-mediated decay (NMD) system, which detects the lack of an open reading frame[14]. Although some asRNAs have been shown to affect translation in human cells[15,16], no cytoplasmic function for asRNAs is currently known in yeast. Thus, the export and cytoplasmic degradation of bulk asRNAs would seem to be a waste of energy for cells. We have discovered that asRNAs act as regulatory RNAs for their sense counterparts through dsRNA formation, which leads to their preferential nuclear export and a subsequent boost in gene expression.

To obtain a global picture of nucleo-cytoplasmic RNA distribution in the eukaryotic model organism *S. cerevisiae*, we fractionated cells and determined the amount of RNA in the cytoplasmic fraction relative to the total RNA by RNA sequencing (RNA-seq) (Fig. 1a, Extended Data Fig. 1a–d and Supplementary Fig. 1a,b). As expected, ribosomal RNAs

(rRNAs) increased in the cytoplasmic fraction, because they are part of the translating ribosomes, whereas small nucleolar RNAs (snoRNAs) were underrepresented, because they function in the nucleus. Interestingly, we found that 23.69% of cellular mRNAs are greatly enriched in the cytoplasm, with many more being nuclear (42.14%). By contrast, 66.33% of asRNAs are greatly enriched in the cytoplasm (Fig. 1b and Extended Data Fig. 1e,f). This finding reveals striking differences in the overall cellular distribution of mRNAs and asRNAs, and led to the surprising observation that bulk asRNAs shuttle out of the nucleus. Furthermore, mRNAs with an equally highly expressed asRNA show on average an equivalent enrichment in the cytoplasmic fraction, whereas mRNAs with no considerable asRNA transcript are more likely to be nuclear (Fig. 1c). This indicates a possible determining role for asRNA transcripts in the cellular localization of their mRNA, perhaps by dsRNA formation.

To identify dsRNAs in yeast we used the dsRNA-specific antibody J2 (ref. 17) in RNA co-immunoprecipitation (RIP) and subsequent RNA sequencing (RIP-seq) experiments (Fig. 1d and Extended Data Fig. 2a–c). This antibody recognizes dsRNAs 40 base pairs (bp) long and has already been used to determine the dsRNA transcriptome of *Escherichia coli*[18]. Remarkably, in yeast, J2 RIPs enriched more than 60% of all asRNAs in the eluate, but only around 13% of the mRNAs (Fig. 1d,e and Extended Data Fig. 2d,e), indicating that most of the asRNAs are part of a double strand, whereas this is only the case for a minority of the sense transcripts. Noticeably, snoRNAs and rRNAs were depleted in the J2 eluate (Fig. 1f), confirming the specificity of the antibody to dsRNAs formed by mRNA and asRNA pairs in yeast. The predominant enrichment of asRNAs is in line with the observation that asRNAs are on average expressed at 10-fold lower levels than the coding transcripts[19] and simply have a higher probability of being present in a dsRNA. Moreover, single-cell analyses have shown that in only 10% of the cells is a sense RNA present together with its corresponding asRNA,

[1]Abteilung für Molekulare Genetik, Institut für Mikrobiologie und Genetik, Göttinger Zentrum für Molekulare Biowissenschaften (GZMB), Georg-August Universität Göttingen, Göttingen, Germany. [2]NGS-Integrative Genomics Core Unit, Institute of Pathology, University Medical Center Göttingen, Göttingen, Germany. ✉e-mail: heike.krebber@biologie.uni-goettingen.de

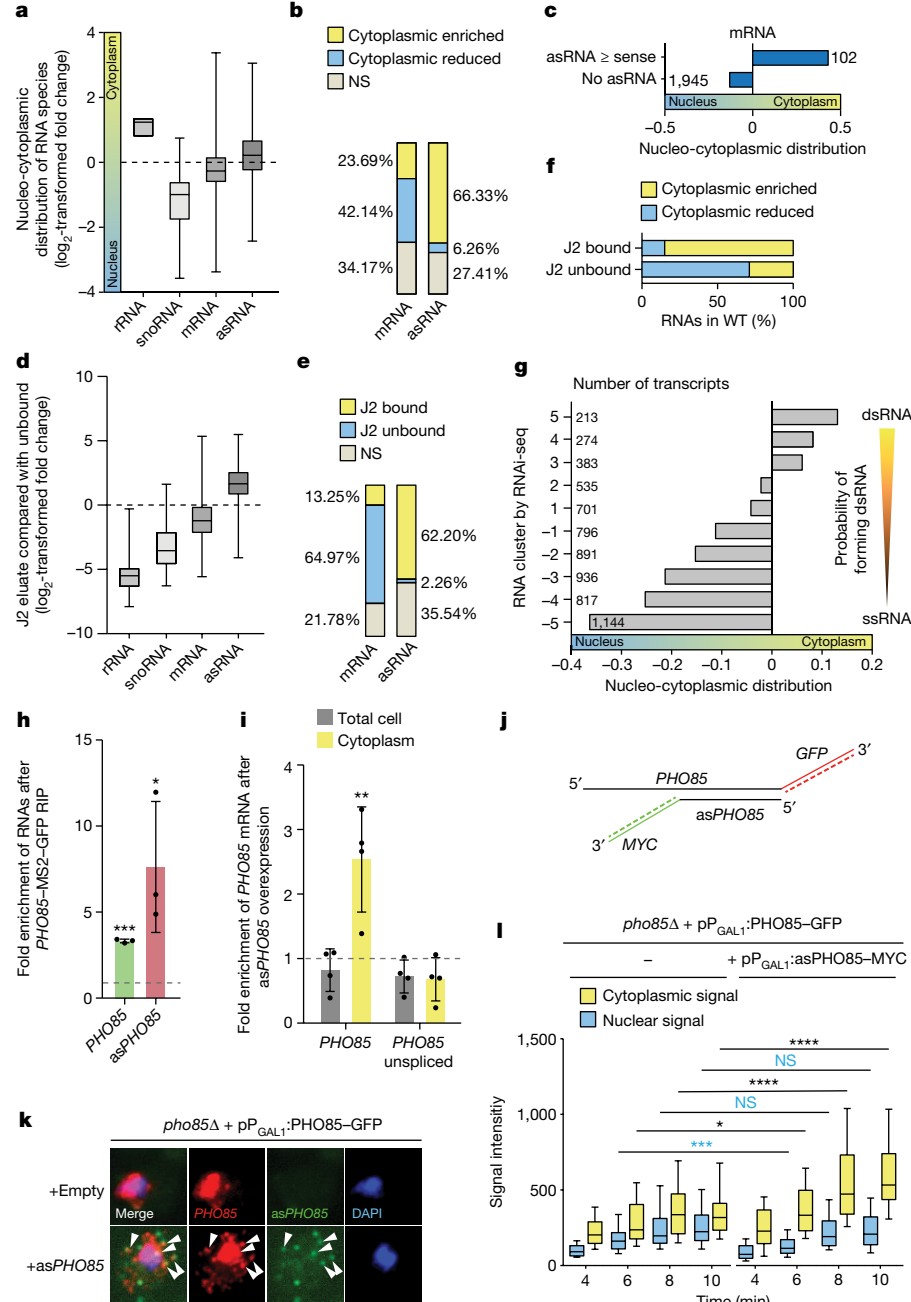

**Fig. 1 | dsRNAs are mainly localized in the cytoplasm. a**, Spatial RNA detection after fractionation RNA-seq experiment. The log₂-transformed fold change of the cytoplasmic fraction compared with total lysate indicates the nucleo-cytoplasmic distribution. From *n* = 3 biologically independent samples. **b**, asRNAs are enriched in the cytoplasm. The percentage of unevenly distributed mRNAs and asRNAs from the fractionation RNA-seq is shown. NS, not significant. **c**, mRNAs localize to the cytoplasm when the asRNA is present in equal or higher amounts. Average nucleo-cytoplasmic distribution of mRNAs, based on their relative asRNA expression. **d**, J2 RIP experiments enriched asRNAs, as determined by the log₂-transformed fold change of the J2 eluate relative to the unbound fraction. From *n* = 3 biologically independent samples. **e**, Percentage of significantly changed transcripts from the J2 RIP-seq. **f**, J2-enriched dsRNAs are cytoplasmic, as determined by their distribution in fractionation RNA-seq. WT, wild type. **g**, RNAs grouped by their enrichment in RNAi-seq[46] analysed for their mean change in fractionation RNA-seq. **h**, *PHO85* mRNA forms dsRNA with *PHO85* asRNA (as*PHO85*) expressed from a plasmid. GFP pull-down with GFP-tagged MS2 loop-binding protein precipitates

MS2-tagged *PHO85* mRNA and co-precipitates *PHO85* asRNA, determined by qPCR. From *n* = 4 biologically independent samples. **i**, Overexpression of *PHO85* asRNA shifts *PHO85* mRNA into the cytoplasm relative to no *PHO85* asRNA (dotted line). The qPCR result after the fractionation experiment and RNA isolation of the total and unspliced *PHO85* transcripts is shown. From *n* = 4 biologically independent samples. **j**, Hybridization with 12 Cy3-labelled probes detects a single *GFP*-tagged *PHO85* mRNA and 15 Alexa647-labelled probes detects the *MYC*(15×) sequence of *PHO85* asRNA. smFISH analysis shows dsRNA formation and a cytoplasmic shift of the *PHO85* mRNA signal with *PHO85* asRNA expression. The *pho85*Δ cells have a galactose-inducible *PHO85*–*GFP* and either an empty plasmid or the galactose-inducible as*PHO85*–*MYC*(15×) plasmid after 8 min galactose induction. The white arrowheads indicate colocalizing signal. From *n* = 4 biologically independent experiments with similar results. **l**, smFISH quantification of nuclear and cytoplasmic *PHO85* signal at different time points after galactose induction with or without simultaneous asRNA induction; *n* > 52 cells over 3 biologically independent experiments.

but around half of the asRNAs are present with their sense transcript (Extended Data Fig. 3a). Most importantly, more than 80% of the RNAs that were enriched in the J2 eluate were cytoplasmic (Fig. 1f), indicating the increased cytoplasmic presence of dsRNAs. By contrast, most of the ssRNAs that did not bind to the J2 antibody were nuclear.

In yeast, dsRNA formation has previously been detected by a screen based on RNA interference (RNAi) in which the dsRNA-degrading RNAi system was artificially established, because *S. cerevisiae* had lost this system during evolution[20,21]. It allowed dsRNAs to be identified after Dicer expression by detection of the accumulating degradation products. The two datasets, the RNAi-seq and our J2 RIP-seq data, show a high correlation (Spearman's rank correlation, *r* = 0.72; Extended Data Fig. 3b). We then grouped RNAs, on the basis of their enrichment in RNAi-seq, into ten clusters (Extended Data Fig. 3c) and applied fractionation RNA-seq. It became apparent that the more an RNA is prone to form a double strand, the more it is likely to localize to the cytoplasm in wild-type cells (*r* = 0.52; Fig. 1g and Extended Data Fig. 3d). Gene-coverage analysis of the RNAi-seq data furthermore revealed that most of these dsRNAs are formed with Xrn1-sensitive unstable transcripts (XUTs), which show greatest cytoplasmic enrichment and longest overlap with their mRNA in lncRNAs, fewer with stable unannotated transcripts (SUTs) and even fewer with cryptic unstable transcripts (CUTs)[22] (Extended Data Fig. 3e–g). Thus, asRNAs, along with mRNAs that have a high probability of being present in a dsRNA with their counterpart, show on average cytoplasmic enrichment.

To investigate whether an asRNA could indeed increase the cytoplasmic localization of its sense transcript, we randomly chose to study the cellular localization of the cyclin-dependent kinase *PHO85* mRNA. Based on its fractionation RNA-seq data, the *PHO85* transcript is expressed with average amounts and is mostly nuclear, whereas its asRNA (*SUT412*) is only barely detectable. First, we validated the double-strand formation of both transcripts in RIP experiments when the asRNA is expressed ectopically from a plasmid. For this, we used a *PHO85* mRNA tagged with 12 MS2 loops and placed the *PHO85* asRNA downstream of a galactose-inducible promoter (*PGAL1*) to control its transcription and raise the expression level. The GFP-tagged MS2-binding protein MCP was co-expressed, subsequently precipitated and the co-purified RNA was analysed. *PHO85* mRNA and its asRNA were present in the eluate, confirming that we had precipitated the RNAs as a double strand (Fig. 1h). Next, we carried out cytoplasmic fractionation experiments while overexpressing *PHO85* asRNA, and these revealed that the *PHO85* mRNA transcript notably shifted its distribution into the cytoplasm through elevated asRNA expression and dsRNA formation (Fig. 1i, Extended Data Fig. 4a,b and Supplementary Fig. 1c,d) For further validation, we placed both transcripts downstream of the inducible *GAL1* promoter, which enabled us to trace their export in single-molecule fluorescence in situ hybridization (smFISH) experiments. Because it is difficult for probes to bind in the dsRNA region, both transcripts were tagged and 12 (for the *GFP* tag) or 15 (for the *MYC* tag) probes with different fluorophores were designed to be complementary to the respective single-stranded tag (Fig. 1j,k). After induction, cells were collected and fixed at different time points for smFISH. The simultaneous induction with and without asRNA resulted in similar signal intensities for the *PHO85* mRNA over the course of the experiment (Fig. 1l). However, in combination with asRNA expression, the cytoplasmic pool of the sense transcript increased substantially compared with no asRNA expression (Extended Data Fig. 4c). The observed colocalization of both transcripts indicates that the effect is due to dsRNA formation. Indeed, a similar observation had already been made in human skin cutaneous melanoma cells, in which an increased transcription of the antisense RNA *TTN-AS1* resulted in increased cytoplasmic localization of its mRNA *TTN*. Remarkably, knockdown of the antisense form reduced tumour growth and metastasis[23]. Notably, allowing dsRNA formation to occur inhibited neither splicing nor nuclear quality control of the *PHO85* mRNA transcript, because no

increased unspliced mRNA was detected in the total lysate or in the cytoplasmic fraction (Fig. 1i). Thus, asRNA can shift the nucleo-cytoplasmic distribution of its mRNA towards the cytoplasm, which is a previously unknown phenomenon that reveals a new layer of gene expression. To investigate whether dsRNA formation might alter the stability of the transcripts, we analysed previously published experimental data[24] from a study that explored RNA stability through a precise pseudo-uridine labelling method and combined that study's data with those of the RNAi-seq-based dsRNA probability and J2 RIP-seq. We observed no correlation between RNA stability and its occurrence in a double strand (*r* = −0.04, *r* = −0.13; Extended Data Fig. 5a,c). Interestingly, mRNAs with a long half-life (*r* = −0.42) and a high expression level (*r* = −0.43) tend to be more nuclear (Extended Data Fig. 5b,c). Importantly, because the relative cytoplasmic distribution is not caused by higher stability of the dsRNA, a faster export rate of dsRNAs might rather be a possible explanation for their increased cytoplasmic presence.

It was already known that mRNAs use multiple molecules of the heterodimer Mex67–Mtr2 (human TAP–p15) for export, and these are recruited during quality control by the guard proteins. The karyopherin Crm1/Xpo1 supports export through interaction with the cap-binding complex attached to the 5′ end of the transcript[1,2,25–28]. It is evident from high-throughput analyses that Mex67 is also implicated in ncRNA export[29]. Furthermore, ncRNAs such as the lncRNA *TLC1* and snRNAs have been shown to be exported via the Mex67, Xpo1 pathway[28,30]. To determine whether dsRNAs also depend on this pathway, we repeated the fractionation RNA-seq in the *mex67-5 xpo1-1* double mutant that was shifted to its restrictive temperature at 37 °C for 1 h (Extended Data Fig. 6a,b). Indeed, all types of RNA were trapped in the nucleus, regardless of their probability of forming dsRNA, demonstrating their dependence on both export factors (Fig. 2a and Extended Data Fig. 7a). Furthermore, immunofluorescence and dot-blot experiments using the J2 antibody confirmed the cytoplasmic localization of dsRNAs in wild-type cells and the nuclear retention in *mex67-5 xpo1-1* (Fig. 2b,c and Extended Data Fig. 7b, c). Importantly, inhibiting translation in the ribosomal subunit export mutant *nmd3-2*, the ribosome subunit joining mutant *rpl10[G161D]* and the inhibitor cycloheximide increased the cytoplasmic dsRNA content, confirming the reliance on translation for resolution (Fig. 2b–d, Extended Data Fig. 7d and Supplementary Fig. 2a). These findings indicate the dependence of dsRNA export on the Mex67 mRNA export pathway and the need for translation to resolve dsRNAs.

To determine whether dsRNAs are exported in preference to ssRNAs, we did an experiment. First, we blocked RNA export in the *mex67-5* mutant by shifting it to 37 °C, causing mislocalization of Mex67 from the nuclear rim to the cytoplasm[1] and in turn nuclear accumulation of ssRNAs and dsRNAs (Fig. 2e). When we lowered the temperature to 25 °C, functional Mex67 returned to the nuclear rim and transport was restored. To see which transcripts reach the ribosome, and thus the cytoplasm, first, we precipitated Rps2 and purified the co-precipitated RNA at specific time points after the export block release (Extended Data Fig. 7e and Supplementary Fig. 2b,c). We found that dsRNAs arrived at ribosomes notably earlier than ssRNAs, providing experimental evidence for the preferential export, and thus translation, of dsRNAs (Fig. 2e). Next, we analysed how faster export induced by asRNA expression would influence the protein and RNA level of the sense gene in quantitative PCR (qPCR) and western blots (Fig. 2f and Supplementary Fig. 2d,e). After *PHO85* asRNA induction, the protein level of Pho85–GFP increased (3.1-fold at 60 min), whereas the level of the *PHO85–GFP* mRNA decreased slightly (0.32-fold at 60 min), because it might no longer be held in the nucleus for potential export. This is an indication of the gene-expression boost of the sense RNA by its asRNA.

The faster export may be due to a different occupancy of the RNAs by export receptors. As suggested previously, the more export proteins cover an RNA, the faster transport is initiated[26,31]. To analyse Mex67 binding to ssRNA and dsRNA, we carried out electrophoretic mobility shift assays (EMSAs). First, we purified recombinant Mex67–Mtr2,

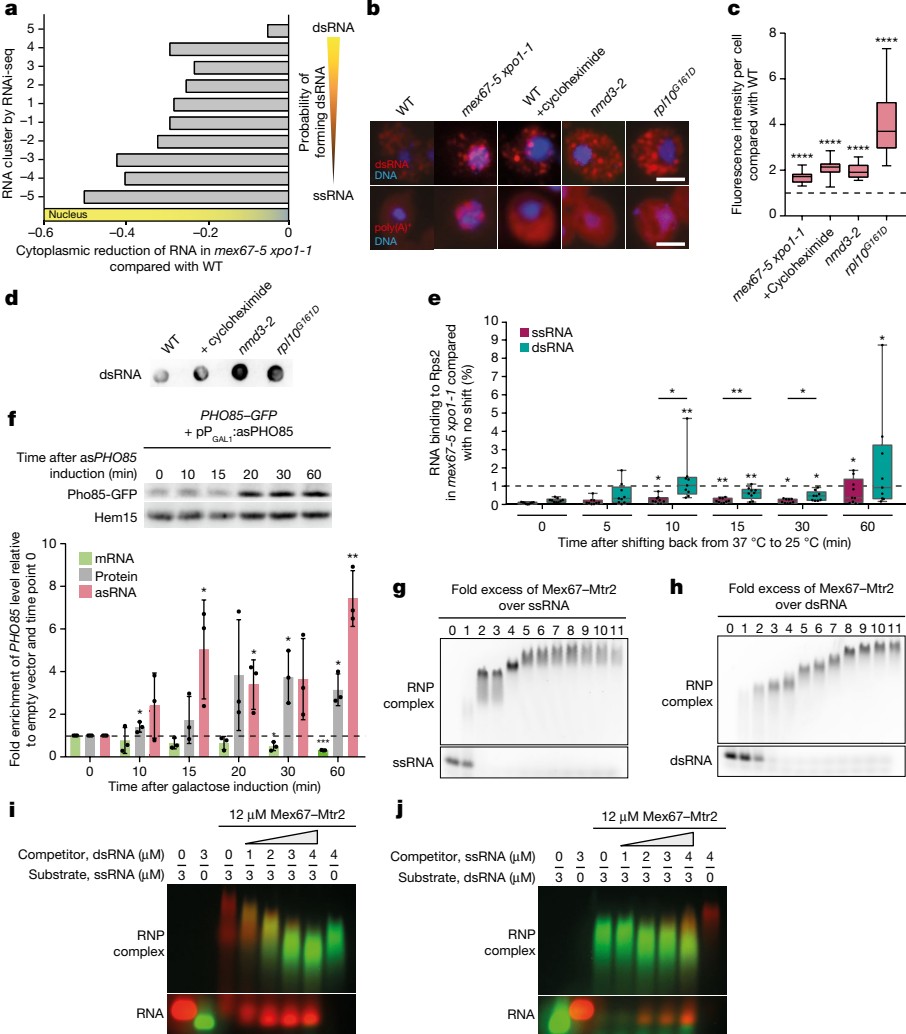

**Fig. 2 | Mex67 preferentially binds dsRNAs for faster nuclear export.**
**a**, dsRNAs accumulate in the nucleus of *mex67-5 xpo1-1*, as determined by fractionation RNA-seq experiment. The average reduction of the cytoplasmic RNA relative to the wild type is based on groups defined by RNAi-seq. From *n* = 3 biologically independent samples. **b**, Exported dsRNAs reach the ribosome. J2 antibody IF with a Cy3-labelled secondary antibody (dsRNA) and FISH with a Cy3-labelled oligonucleotide d(T) probe were done for 1 h at 37 °C. From *n* = 3 biologically independent experiments. Scale bars, 3 µm. **c**, Fluorescence intensity quantification in J2 IF for 1 h at 37 °C for each condition shown in **b**; *n* > 20 cells over 3 biologically independent experiments. From left to right: *P* = 1.27 × 10⁻⁸, *P* = 3.03 × 10⁻²⁵, *P* = 1.98 × 10⁻²³ and *P* = 2.39 × 10⁻¹⁸. **d**, dsRNA levels increase when translation is inhibited. Dot-blots with RNA from the indicated strains and treatments after 1 h at 37 °C were detected with the J2 antibody. From *n* = 3 biologically independent experiments with similar results. **e**, dsRNAs contact ribosomes before ssRNAs. We shifted *mex67-5* to 37 °C for 1 h to block

RNA export. It was subsequently released by lowering the temperature to 25 °C. RIP experiments with Rps2–GFP from different time points were done. Three ssRNA and three dsRNA targets were analysed by qPCR. From *n* = 3 biologically independent experiments. **f**, Induction of *PHO85* asRNA expression changes both protein and mRNA levels of *PHO85*, as shown by western blot quantification and qPCR. Hem15 served as loading control and for normalization. From *n* = 3 biologically independent experiments. **g**,**h**, More Mex67 molecules can bind to dsRNA than to ssRNA. EMSA with FAM-labelled ssRNAs (**g**) or dsRNAs (**h**) was carried out by adding increasing amounts of recombinant TAP-tagged Mex67–Mtr2. From *n* = 3 independent experiments with similar results. **i**,**j**, Competition assay detects preferential Mex67–Mtr2 binding to dsRNA. Cy5-labelled ssRNAs (red) were pre-incubated with Mex67–Mtr2. Subsequently, increasing amounts of a FAM-labelled dsRNA (green) were added as a competitor (**i**). Increasing amounts of Cy5-labelled ssRNA were added to pre-bound FAM-labelled dsRNA (**j**). From *n* = 3 independent experiments with similar results.

which was shown to bind directly to mRNAs[32]. Mex67–Mtr2 was added in varying molar excess to fluorescein amidite (FAM)-labelled 36-nucleotide-long ssRNA or dsRNA (Extended Data Fig. 7f and Supplementary Fig. 4a–c) and incubated for 15 min before the samples were analysed on gel electrophoresis. A 2-molar excess of Mex67–Mtr2 fully upshifted ssRNA, which was saturated at 5-molar excess (Fig. 2g and Supplementary Fig. 3a). By contrast, dsRNA reached binding saturation at a 10-molar excess, with a full upshift at a 3-molar excess (Fig. 2h and Supplementary Fig. 3b). This indicates that dsRNAs can bind about two heterodimers every 7 bp, whereas ssRNAs accommodate only one in the same length. Subsequently, we carried out competition assays in which we added either FAM-labelled dsRNA or Cy5-labelled ssRNA

to a Mex67–Mtr2 pre-formed complex on similarly labelled dsRNA or ssRNA, respectively. Mex67–Mtr2 dissociated from ssRNA and associated with dsRNA in the presence of increasing dsRNA amounts, but it showed almost no displacement properties when ssRNA was titrated against the dsRNA complex (Fig. 2i,j and Supplementary Fig. 3c,d). Switching the labels made no difference and resulted in the same effect (Extended Data Fig. 7g,h and Supplementary Fig. 4d,e). These findings show that Mex67–Mtr2 binds preferentially and more extensively to dsRNA, explaining its preference in export.

Our findings indicate that asRNAs can boost the expression of individual transcripts. The broad existence of annotated asRNAs in the yeast genome indicates a general usage of asRNA boosts to navigate

gene expression, which would explain their prevalence. It is likely that not all boosting asRNAs have been discovered yet, because they may be transcribed only under specific conditions to steer the cells in a particular direction, such as during development or stress. For instance, a previous study found a new set of asRNAs that are suppressed by the chromatin modifier Set2 (ref. 33). These down-regulated transcripts, termed Set2-repressed antisense transcripts (SRATs), are antisense to stress response and ageing-related transcripts. To determine whether these pairs could form dsRNAs, we carried out J2 immunofluorescence (IF), J2-IP and dot-blot experiments in *set2Δ* and found a significant increase in the total amount of dsRNA in the cytoplasm (Extended Data Fig. 8a–c,e and Supplementary Fig. 5a), indicating that the induced SRATs form double strands with their sense counterparts and are exported as dsRNA. Cell fractionation experiments and qPCRs with a Set2-responsive transcript, *SEG2* asRNA, confirmed this observation and showed an up-regulation of the asRNA when *SET2* was deleted (Extended Data Fig. 8d), resulting in dsRNA formation (Extended Data Fig. 8e) and a subsequent increased presence of *SEG2* mRNAs in the cytoplasm (Extended Data Fig. 8f,g and Supplementary Fig. 5b).

Changes to the cellular expression programs are sometimes necessary, for instance for development, during ageing and in response to stress. Each situation might generate new asRNA transcripts that boost the expression of individual mRNAs. Analysis of previously published RNA-seq data that were generated after osmotic shock[34] revealed not only that stress-responsive mRNAs increased during periods of stress, but so did asRNAs[13] (Fig. 3a). Indeed, 95% of significantly increased mRNAs with significantly changed asRNA show an up-regulation of the asRNA (Fig. 3b). To visualize the transcriptome-wide changes to the expression program, we determined the dsRNA formation in cells shocked with salt (0.7 M NaCl) or ethanol (10% EtOH) through J2-IF experiments (Fig. 3c,d). Although bulk mRNA accumulates in the nucleus and dissociates from Mex67 under stress conditions[32], the newly produced dsRNA reached the cytoplasm in the first 5 min. These dsRNAs represent the preferentially exported stress-induced mRNAs that hybridized with their asRNA.

If this mechanism of regulated, boosted gene expression resulting from dsRNA formation were of a general nature, the degradation of dsRNA should be hazardous to cells. Previous work has shown that double strands of mRNA and asRNA are degraded by RNaseIII from *E. coli*[18]. In *S. cerevisiae*, such studies found unexplained toxicity of cells expressing RNaseIII in growth analyses that could not be attributed to the degradation of rRNA or general mRNA[35], but it might be caused by unrecognized dsRNA degradation. To confirm the specificity of RNaseIII for dsRNA, we exposed isolated total RNA to recombinant RNaseIII and subsequently detected the remaining dsRNA by J2 dot-blot. After RNaseIII digestion, the dsRNA was noticeably reduced (Extended Data Fig. 8h and Supplementary Fig. 5c). To analyse its effects on living cells, we expressed RNaseIII in vivo from the inducible *GAL1* promoter and directed the protein to different compartments. Tagging the enzyme with a nuclear localization signal resulted in cell death (Fig. 3e). Expressing RNaseIII with both a nuclear localization signal and a nuclear export signal (NES) was equally toxic. This fusion protein can shuttle between the nucleus and the cytoplasm and was visible mostly at the nuclear rim (Fig. 3f). Interestingly, a construct that restricted the dsRNA-degrading enzyme to the cytoplasm (RNaseIII–NES) was tolerated in rich medium. This may be because the cytoplasmic dsRNA is somehow protected or because the subsequent translation is quite fast. The possibility that the cytoplasmic construct is not functional is unlikely because reduced dsRNA levels were detectable in the translation-defective strain *rpl10^{G161D}* (Extended Data Fig. 8n and Supplementary Fig. 5d), in which the cytoplasmic dsRNA accumulates (Fig. 2b). Most importantly, although the cytoplasmic RNaseIII–NES was tolerated in rich medium, its expression during osmotic stress resulted in severe growth defects and dsRNA reduction (Fig. 3g,h). This was also the case for the cytoplasmic-operating RNAi system.

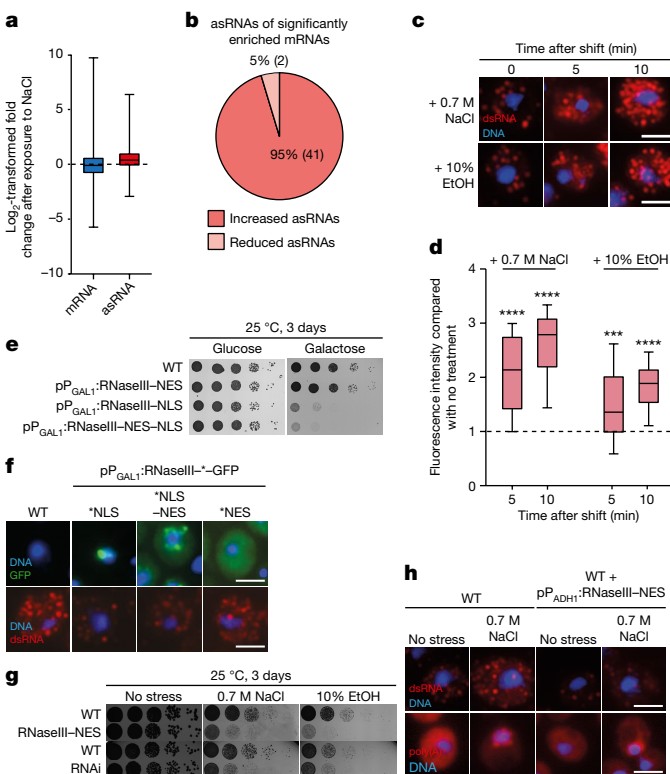

**Fig. 3 | dsRNA formation is essential for cells changing their expression program. a**, Stress increases asRNA levels. A genome-wide RNA analysis of cells incubated with 0.6 M NaCl (ref. 34) for 30 min was used to find changes in sense mRNA and asRNA expression compared with unstressed conditions, and is shown here by the $\log_2$-transformed fold change. From $n = 2$ biological independent samples. **b**, Stress-responsive mRNAs are accompanied by increased asRNA expression. Significantly changed asRNAs of significantly increased mRNAs from the RNA-seq[34] are shown after 30 min exposure to 0.6 M NaCl. **c**, The amount of dsRNA increases under stress conditions. J2-IF is shown for the wild type exposed to the indicated stress conditions. From $n = 3$ biologically independent experiments. **d**, Quantification of the J2-IF displayed in **e**. From $n = 30$ cells over 3 independent experiments. Left to right: $P = 6.91 \times 10^{-9}$, $P = 1.59 \times 10^{-15}$, $P = 7.57 \times 10^{-4}$, $P = 5.47 \times 10^{-12}$. **e**, dsRNA degradation by the bacterial ribonuclease RNaseIII in the nucleus of yeast cells is lethal. We spotted 10-fold serial dilutions of the wild type containing the indicated plasmids onto glucose (no induction) or galactose (with induction) plates and incubated for 3 days. From $n = 3$ biologically independent experiments with similar results. **f**, J2-IF and localization of the GFP- and transport signal-tagged RNaseIII fusion proteins in yeast cells. Plasmid-containing wild-type cells were grown to the logarithmic phase before the RNaseIII expression was induced by adding galactose. From $n = 3$ biologically independent experiments with similar results. **g**, Cytoplasmic RNaseIII is not tolerated in cellular stress situations. We spotted 10-fold serial dilutions of wild-type cells containing either a constitutively expressed RNaseIII–NES from the *ADH1* promoter or the RNAi system onto plates and incubated for 3 days at 25 °C. From $n = 3$ biologically independent experiments with similar results. **h**, Stress-induced dsRNA is degraded by cytoplasmic RNaseIII. J2-IF and oligonucleotide d(T) FISH are shown either without stress or after 30 min incubation with 0.7 M NaCl. From $n = 3$ biologically independent experiments. All scale bars, 3 μm.

Together, these data indicate that the asRNA boost of mRNA expression through dsRNA formation is particularly important in changing and challenging situations.

An enzyme that mediates dsRNA formation is most likely to be an RNA helicase, because these enzymes are known not only for their RNA unwinding but also for their dsRNA-binding, dsRNA-annealing and dsRNA-clamp activities[36,37]. Interestingly, deletions of the two nuclear helicases Dbp2 or Mtr4 have been reported to increase XUT-asRNA levels[38]. Mtr4 is an enzyme known to be involved in RNA degradation[39],

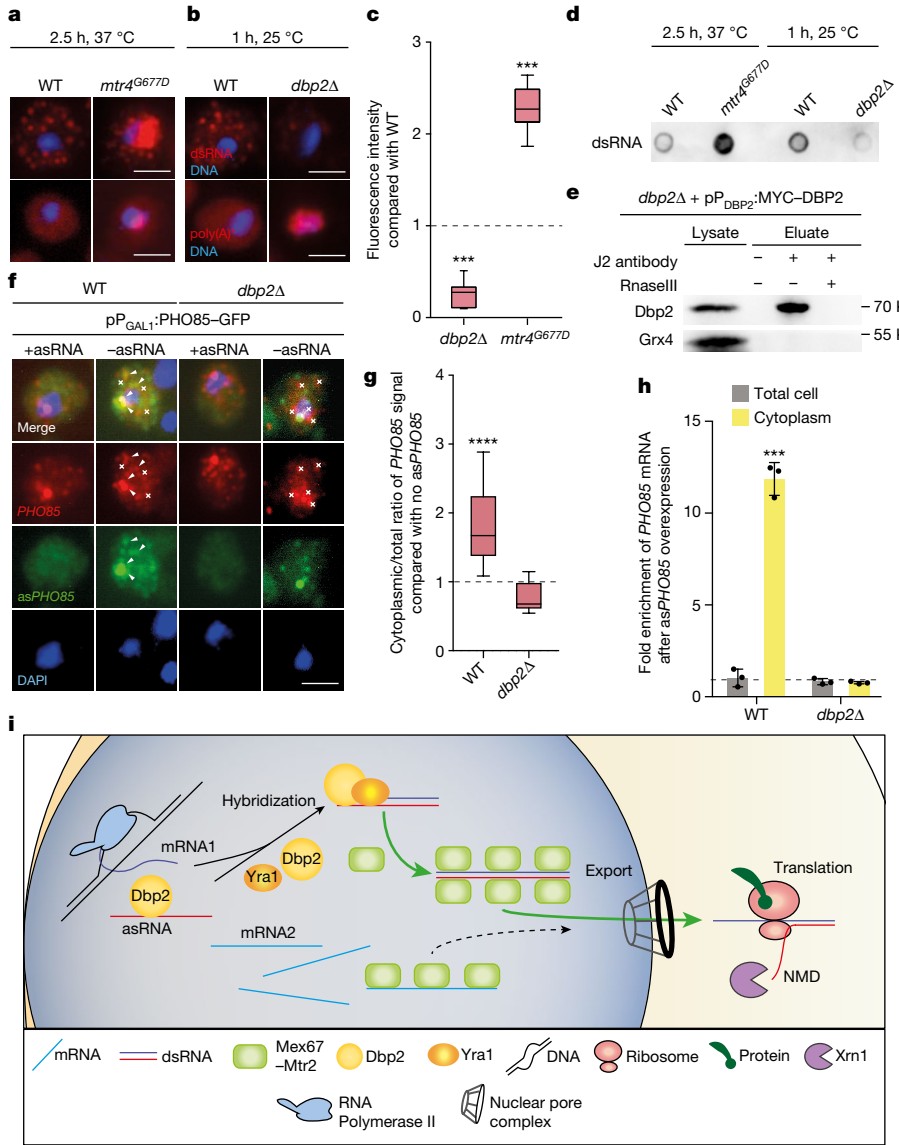

**Fig. 4 | Dbp2 induces dsRNA formation. a**, We found that dsRNAs accumulate in the nucleus of $mtr4^{G677D}$ at 37 °C. J2-IF and oligonucleotide d(T)-FISH are shown. **b**, Disturbed dsRNA formation in $dbp2\Delta$. Strains were changed to the non-permissive temperature for $dbp2\Delta$ of 25 °C. **c**, Signal quantification from **a** and **b**. From $n > 40$ cells over 3 biologically independent experiments. Left to right, $P = 7.83 \times 10^{-35}$, $P = 8.56 \times 10^{-34}$. **d**, J2 dot-blot of isolated RNA. From $n = 3$ biologically independent experiments with similar results. **e**, Dbp2 binds to dsRNA. Western blot of J2 Co-IP from cells expressing $MYC$-tagged $DBP2$ with or without the addition of recombinant RNaseIII. Grx4 is negative control. From $n = 3$ biologically independent replicates with similar results. **f**, dsRNA formation and cytoplasmic shift of $PHO85$ mRNA after $PHO85$ asRNA expression was lost in $dbp2\Delta$. Wild type and $dbp2\Delta$ carrying the galactose-inducible $PHO85$–$GFP$ plasmid and either an empty vector or the galactose-inducible as$PHO85$–$MYC$(15×) plasmid were used for smFISH after 8 min galactose induction. From $n = 3$ biologically independent experiments. **g**, Quantification of smFISH for the cytoplasmic/total signal ratio of $PHO85$ mRNA either with or without (dotted line) simultaneous asRNA expression. From $n > 23$ cells examined over 3 biologically independent experiments. **h**, The increased presence of $PHO85$ mRNA in the cytoplasm after $PHO85$ asRNA overexpression was abolished in $dbp2\Delta$. After the induction of $PHO85$ asRNA, cells were shifted to 25 °C for 1 h before cytoplasmic fractionation, RNA isolation and qPCR. From $n = 3$ biologically independent samples. **i**, Model for the preferential export of dsRNAs. After transcription, ssRNAs are eventually bound by Mex67, leading to low-level export and translation in the cytoplasm. Gene expression is boosted by the transcription of asRNA and subsequent dsRNA formation of sense–antisense pairs by the helicase Dbp2 and its co-factor Yra1. The dsRNA preferentially binds to Mex67 for nuclear export and mRNA translation. Ribosomes recognize the non-coding property of the asRNAs and subsequent NMD-mediated degradation. This mechanism ensures preferential gene expression. All scale bars, 3 μm.

whereas for Dbp2, strand annealing under the mediation of Yra1 has already been demonstrated in vitro and has been suggested to be important for messenger ribonucleoprotein (mRNP) assembly[40,41]. Furthermore, Yra1 was shown to recruit Mex67 for mRNA export[42]. To investigate whether Dbp2 and/or Mtr4 might be involved in dsRNA biogenesis, we used a cold-sensitive $DBP2$-knockout strain[41] (Extended Data Fig. 9a) and the temperature-sensitive $mtr4^{G677D}$ mutant[43] in J2 IF studies and J2 dot-blot experiments. Interestingly, the $mtr4$ mutant accumulated dsRNA in the nucleus, indicating that this helicase is not

required for dsRNA formation but instead for its decay after quality control (Fig. 4a,c,d, Extended Data Fig. 9b and Supplementary Fig. 6a). Most importantly, however, the absence of Dbp2 resulted in a clear decrease in dsRNA formation and a simultaneous nuclear accumulation of poly(A)$^+$ RNA (Fig. 4b–d and Extended Data Fig. 9c), identifying Dbp2 as the dsRNA-forming helicase in vivo and showing its mRNA export-supporting function. Indeed, it has been shown that the sense and corresponding antisense transcripts changed by the deletion of $DBP2$ strongly correlate[44], which supports our findings. Thus,

we reconsidered previously published high-throughput analyses of *dbp2*, in which RNA secondary structures were determined through the labelling of free nucleotides with dimethyl sulfate in the wild type and *dbp2Δ* (ref. 45). We applied this dataset to the RNAi-seq data[21] and found that the more an RNA is prone to be in a double strand, the more nucleotides became accessible and were labelled in the *dbp2Δ* compared with the wild type (Extended Data Fig. 9d). Interestingly, individual-nucleotide resolution cross-linking and immunoprecipitation sequencing (iCLIP-seq) data[45] showed similar binding of Dbp2 to all transcripts when analysed in the context of RNAi-seq[21] (Extended Data Fig. 9e), indicating that the helicase contacts all transcripts initially, possibly for unwinding as previously suggested[45]. The potential for subsequent dsRNA formation then depends on the availability of the respective asRNA and on the contact with Yra1.

To find further experimental evidence, we first confirmed that Dbp2 binds to dsRNA in vivo through J2-IP (Fig. 4e and Supplementary Fig. 6b). This binding was lost in the presence of recombinant RNaseIII. To validate the in vivo relevance for the dsRNA-formation function of Dbp2, we repeated the experiment from Fig. 1i–k in the absence of Dbp2. We found that although the asRNA was still highly enriched in *dbp2Δ* after galactose induction, it was not able to manipulate the localization of its sense *PHO85* mRNA (Fig. 4f–h, Extended Data Fig. 9f,g and Supplementary Fig. 6c,d). This finding indicates that Dbp2 is the key factor for dsRNA formation that enables the preferential export of dsRNAs.

In conclusion, our findings reveal a new layer of regulated gene expression. Boosting asRNAs anneal with their sense counterpart through Dbp2-mediated dsRNA formation. These dsRNAs are preferentially exported and subsequently the respective sense transcripts are preferentially expressed (Fig. 4i). This mechanism is particularly important for effective cellular adaptation and adds preferential export as a new layer of regulated gene expression. Furthermore, it could also explain how pervasive transcription controls gene expression, and why so many asRNAs are generated and travel into the cytoplasm.

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

## Methods

### Yeast strains, plasmids and oligonucleotides

All the yeast strains used in this study are listed in Supplementary Table 1 and the plasmids are listed in Supplementary Table 2. Strains were cultivated and grown in standard medium at 25 °C. The diploid strain HKY2065 was sporulated and subjected to tetrad dissection followed by analysis of haploid spores for their genetic markers.

### FISH

The experiments were essentially carried out as previously described[47]. To detect poly(A)$^+$ RNA, a Cy3-labelled oligonucleotide d(T)$_{18}$ probe (Sigma) was used. Cells were grown to mid-logarithmic phase (around $1 \times 10^7$ cells per ml) before they were treated as indicated. Cells were fixed by adding formaldehyde to a final concentration of 3.7% for 40 min at room temperature. After washing, permeabilization and pre-hybridization, the Cy3-labelled d(T)$_{18}$ probes were added and hybridized overnight at 37 °C. DNA was stained with DAPI (Sigma) for 2 min. Microscopy studies were carried out using a Leica AF6000 microscope and an HCX PL APO CS ×63 objective lens. Pictures were obtained using a LEICA DFC360FX camera with a resolution of 1,392 × 1,040 px and LAS AF 2.7.3.9 software (Leica). Images were quantified using Fiji software.

### smFISH

The experiment was conducted largely as described above. Cells were grown in 2% raffinose to the logarithmic-growth phase. The expression of *PHO85* mRNA and its asRNA were induced by adding 2% galactose. Cells were collected after the indicated times and fixed for 20 min in 3.7% formaldehyde. The probes used are listed in Supplementary Table 5. They were incubated for 3 h at 37 °C. Thereafter, the washing steps with SSC were carried out for 15 min each, as described in the FISH protocol. Quantification of the signal was carried out using Fiji software. To determine the nuclear signal, the DAPI signal was used as a reference. The boundary of the total cell was determined using Nomarski optic. The cytoplasmic signal was calculated by subtracting the nuclear signal from the total signal. The background signal was measured three times per image and subtracted from the measured signal of the cell as follows: integrated density − (selected area) × (mean fluorescence of background readings), which resulted in the final signal strength that was used for all images. For every time point, cells from three independent biological repetitions were quantified.

### Immunofluorescence

Cells were grown, collected and treated as described for the FISH experiment. After permeabilization, cells were blocked in ABB (0.1 M Tris, pH 9.0, 0.2 M NaCl, 5% FCS, 0.3% Tween, 500 µg ml$^{-1}$ transfer RNA) and incubated for 1 h at 37 °C, followed by ABB with the addition of 1/200 µl of the J2 antibody (1 µg µl$^{-1}$) from Scicons[17] and 0.2% Triton for 2 h at 37 °C. The addition of Triton prevented binding to membrane-bound glycan RNA, which was already known to be an antigen of the J2 antibody. Subsequently, cells were washed with 0.5% Triton in 1× PBS for 15 min, twice with 1× PBS for 15 min and finally with ABB for 30 min. The secondary Cy3-conjugated anti-mouse antibody in ABB (1:200) was thereafter incubated for 1 h at room temperature. Subsequently, cells were washed with 0.5% Tween in 1× PBS for 10 min and twice in 1× PBS for 10 min. Nuclei were stained with DAPI (Sigma) and mounting, microscopy and quantification were carried out as described in the FISH experiment.

### GFP microscopy

The visualization of GFP-tagged proteins in vivo was essentially done as previously described[47]. Cells were grown in glucose (2%)-containing medium until the early logarithmic phase ($0.5 \times 10^7$ cells per ml), washed once with 1 ml sterile H$_2$O, transferred into galactose (2%)-containing medium and grown for 6 h to induce the expression of RNaseIII constructs. Next, cells were fixed with 3.7% formaldehyde for 1 min at room temperature and washed twice with 1 ml P-Solution (0.1 M potassium phosphate buffer, pH 6.5, 1.2 M sorbitol) before adding 20 µl on a polylysine-coated slide for 15 min at room temperature. Permeabilization, DNA staining, microscopy and quantification were carried out as described in the FISH experiment.

### Cytoplasmic fractionation

To detect RNAs in the cytoplasm, cells were grown to mid-logarithmic phase ($2 \times 10^7$ cells per ml), washed once with 1 ml YPD/1 M sorbitol/2 mM DTT and resuspended in YPD/1 M sorbitol/1 mM DTT with the addition of zymolyase (100 mg ml$^{-1}$) to spheroplast cells. Before cytoplasmic fractionation, 200 µl of the cell suspension was taken for total lysate control. For the analysis shown in Fig. 1a and similar work, after spheroblasting, cells were diluted in 50 ml YPD/1 M sorbitol for 30 min at 25 °C before they were shifted to 37 °C for 1 h. After shifting, 10 ml was taken for total cell lysis. Next, cells were cooled on ice and centrifuged for 5 min at 2,000 rpm. For cytoplasmic fractionation, the cell pellets were resuspended in 500 µl Ficoll buffer (18% Ficoll 400, 10 mM HEPES, pH 6.0) and cells were lysed by adding 1 ml buffer A (50 mM NaCl, 1 mM MgCl$_2$, 10 mM HEPES, pH 6.0) and 1 µl Ribolock RNase Inhibitor (Thermo Fisher). The suspension was vortexed and centrifuged for 10 min at 2,000 rpm. The resulting supernatant reflects the cytoplasmic fraction. To verify correct fractionation, samples were analysed in western blots for the presence of the cytoplasmic Zwf1 (anti-Zwf1 in TBS-T (50 mM Tris-HCl, pH 7.4, 150 mM NaCl, 0.1% Tween 20), 1:4,000) and the nuclear proteins Yra1 (anti-Yra1 in TBS-T, 1:1,000) and Nop1 (anti-Nop1 in TBS-T, 1:4,000). RNA was isolated using a Nucleo-Spin RNA Kit (Macherey and Nagel).

### J2 RNA co-immunoprecipitation experiment

Yeast strains were grown to mid-logarithmic phase ($2 \times 10^7$ cells per m) followed by ultraviolet cross-linking with a wavelength of 254 nm for 7 min. Cells were collected and lysed in RIP buffer (25 mM Tris-HCl, pH 7.5, 150 mM NaCl, 2 mM MgCl$_2$, 0.5% (v/v) Triton X-100, 0.2 mM PMSF, 0.5 mM DTT, 10 U RiboLock RNase Inhibitor (Thermo Fisher) and protease inhibitor (Roche)) by using a FastPrep-24 machine (MP Biomedicals) with shaking three times for 30 s at 5.5 m s$^{-1}$. After centrifugation, 30 µl of the supernatant was taken for input control and the remaining lysate was incubated with or without 3 µl of the J2 antibody (1 µg µl$^{-1}$)[17] from Scicons and the addition of recombinant ShortCut RNaseIII (NEB) for 30 min at 4 °C. After the first incubation, the lysates were transferred to prewashed G-sepharose beads and incubated for another 90 min at 4 °C. The beads were then washed five times with RIP buffer (0.25% Triton). The supernatant was removed and SDS loading dye (125 mM Tris-HCl, pH 6.8, 2% SDS, 10% glycerol, 5% 2-mercaptoethanol, bromphenolblue) was added. Subsequently, samples were incubated at 95 °C for 5 min and loaded onto an SDS gel followed by western blotting and staining with MYC (anti-MYC in TBS-T, 1:,) and Grx4 (anti-Grx4 in TBS-T, 1:4,000).

### J2 RIP for RNA-seq

All yeast strains were grown to mid-logarithmic phase ($2 \times 10^7$ cells per ml). Total RNA was isolated with TRIzol reagent. After the first ethanol precipitation, a DNaseI treatment was conducted followed by a second precipitation overnight. The obtained RNA was eluted in RNase-free water. Then 90 µg RNA and 3 µg J2 antibody were incubated in 500 µl PBST for 120 min at 4 °C (1× PBS, 0.5% Tween-20). After the first incubation, the RNA–antibody mix was transferred to prewashed G-sepharose beads and incubated for another 120 min at 4 °C. The beads were centrifuged for 1 min at 4,000 rpm at 4 °C. The supernatant was transferred a second time, together with 3 µg J2 antibody, to freshly washed beads and incubated for 120 min. Subsequently, these beads were centrifuged and the supernatant was used as the unbound fraction. The beads from the first incubation were washed five times with 1 ml PBST. Between each

step, the beads were centrifuged for 1 min at 4,000 rpm at 4 °C. Finally, the RNA was purified from the unbound fraction and from the eluates by TRIzol-chloroform (Ambion RNA by Life Technologies) extraction and forwarded to RNA sequencing. We repeated the experiment three times, and it showed a high reproducibility.

### J2 dot-blot

Cells were grown to the logarithmic-phase and shifted, if necessary, as indicated. RNA isolation was carried out with TRIzol reagent. Then 1 µg of the isolated RNA was applied onto a nylon membrane, which was blocked in PBST (1× PBS, 1% Tween-20), 0.05 mg ml$^{-1}$ ssDNA and 5% (w/v) non-fat dried milk. Subsequently, the J2 antibody (anti-dsRNA in PBST, 1:5,000) was added and incubated for 2 h at room temperature. Finally, there were two washing steps with PBST, each for 15 min at room temperature, before the HRP-coupled goat anti-mouse secondary antibody was added in PBST for 1 h. Finally, the membrane was washed again three times with PBST for 10 min at room temperature, before the ECL detection was carried out with a Fusion FX7 Edge 18.06c (Vilber). Quantification was finalized with the analysis software Bio-1D from Vilber Lourmat.

### Protein isolation and purification

Transformed Rosetta 2 *E. coli* cells were grown in 200 ml LB medium with ampicillin (100 µg ml$^{-1}$) and chloramphenicol (25 µg ml$^{-1}$) overnight, diluted to OD$_{600}$ = 0.1 in 1,200 ml Terrific Broth medium (28.8 g yeast extract, 24 g Trypton, 9 ml 50% glycerin, 17 mM KH$_2$PO$_4$, 72 mM K$_2$HPO$_4$) and 100 µg ml$^{-1}$ ampicillin. The diluted cells were incubated at 32 °C and 130 rpm for 3 h, followed by 37 °C and 130 rpm for 1 h. For protein induction, 1.2 ml of 1 M IPTG was added and the culture was further incubated at 16 °C and 130 rpm overnight. After induction, cells were washed in 200 ml IMAC loading buffer (50 mM NaH$_2$PO$_4$, 500 mM NaCl, 10 mM Imidazol, pH 7.8) and finally resuspended in 75 ml IMAC loading buffer with Roche complete protease inhibitor (one tablet per 50 ml). Cells were lysed using a microfluidizer with the setting 3 times at 700 bar. Thereafter, the lysate was centrifuged at 15,000$g$ for 90 min. Cleared lysate was loaded onto a 5 ml HisFF column and subsequently washed with IMAC exchange buffer, then 1 M LiCl, again with IMAC exchange buffer, and finally with IMAC loading buffer. The proteins were eluted with IMAC elution buffer (50 mM NaH$_2$PO$_4$, 500 mM NaCl, 400 mM Imidazol, pH 7.8) and dialysed against heparin base buffer (40 mM HEPES KOH, 100 mM KCl, pH 7.5) overnight. After dialysis, the eluate was loaded onto a heparin column and again eluted with heparin elution buffer (40 mM HEPES-KOH, 100 mM KCl, 2 M NaCl, pH 7.5). Finally, the eluate was dialysed in dialysis buffer (30 mM HEPES-KOH, 160 mM KCl, pH 7.6) for 2 days. Protein concentration was determined by measuring the optical density at 280 nm.

### EMSA

Either ordered FAM or Cy5-labelled RNAs (Sigma Aldrich) were used. Every RNA contained 36 nucleotides and had the same amount of C, G, T and A (Supplementary Table 4). dsRNAs were formed by incubating 20 µM of the labelled and 20 µM of the reverse complementary non-labelled RNA in dialysis buffer (30 mM HEPES-KOH, 160 mM KCl, pH 7.6) at 65 °C for 5 min and immediate subsequent cooling on ice. Next, 4 µM dsRNAs or ssRNAs were incubated with purified Mex67–Mtr2 and 2 µl Ribolock RNase Inhibitor (Thermo Fisher) in dialysis buffer, resulting in a final volume of 20 µl, at 30 °C for 15 min. For Fig. 2c, Mex67–Mtr2 was added in increasing amounts from 4 µM to 44 µM. For the competition assay depicted in Fig. 2d, 12 µM Mex67–Mtr2 was added to 3 µM substrate RNA, resulting in a molar ratio of 1:4 between substrate RNA and Mex67. The competitor RNA was added after the first incubation and further incubated at 30 °C for 15 min. Finally, a 6× loading dye (10 mM Tris-HCl, pH 7.6, 60% glycerol, 60 mM EDTA, 0.03% bromophenol blue) was added and the samples were loaded onto a 0.5% agarose gel with 1× TAE (40 mM Tris, 1 mM EDTA, 20 mM acetic acid, pH 9.5) running in 1× TAE, pH 9.5. Complexes were separated by running native gels at 300 V and 4 °C for 40 min. In-gel detection was carried out with a Fusion FX7 Edge 18.06c (Vilber) using the filter F-595 YR and Epi-Light module C530, or filter F-710 and Epi-Light module C640, with Evolution-Capt. Edge software.

### Export release assay

The *mex67-5 xpo1-1* RPS3-GFP strain was grown to the mid-logarithmic phase (2 × 10$^7$ cells per ml) and shifted to 37 °C for 2 h. Cells were collected either directly after shifting (0 min) or after shifting them back to 25 °C for 5 min, 10 min, 15 min, 30 min and 60 min. The cell pellets were frozen in liquid nitrogen and subsequent RIP experiments were carried out as described for the J2 RNA co-immunoprecipitation experiments, with the exception that GFP Trap beads were used and no antibody was added. After the final washing step, the beads were split in half for RNA isolation with TRIzol reagent and subsequent qPCRs and for SDS-PAGE and western blot analysis of GFP (anti-GFP in TBS-T, 1:4,000) and Aco1 (anti-Aco1 in TBS-T, 1:2,000). For qPCR measurements, the ssRNAs *RPS17A*, *RPS6A* and *TDH1* and the dsRNAs *FRE5*, *HPF1* and *PRY3* were analysed. dsRNA targets were chosen using three criteria: the asRNA had a higher RPKM (reads per kilobase million) than the sense RNA; they were identified as dsRNA in an RNAi-seq experiment[21]; and they are enriched in J2 RNA-seq. The ssRNA targets were chosen because of the opposed criteria: the level of the asRNA is less than 1:100 compared with the mRNA and they are not enriched in either RNAi-seq nor J2-seq.

### Cell lysis for protein and RNA quantification

Cells (30 ml) were grown overnight in synthetic medium containing 2% raffinose until the logarithmic phase. For asRNA induction, 2% galactose was added. At each indicated time point, a sample of 5 ml was taken and centrifuged at 4,000 rpm and 4 °C for 4 min. Cells were lysed in 400 µl RIP buffer (25 mM Tris-HCl, pH 7.5, 150 mM NaCl, 2 mM MgCl$_2$ 0.5% (v/v) Triton X-100, 0.2 mM PMSF, 0.5 mM DTT, 10 U RiboLock RNase inhibitor (Thermo Fisher) and protease inhibitor (Roche)) and divided into two samples. SDS loading dye (125 mM Tris-HCl, pH 6.8, 2% SDS, 10% glycerol, 5% 2-mercaptoethanol, bromphenolblue) was added to one of the samples. Subsequently, samples were incubated at 95 °C for 5 min and loaded onto an SDS gel followed by western blotting and staining of GFP (anti-GFP in TBS-T, 1:4,000) and Hem15 (anti-Hem15 in TBS-T, 1:5,000). The RNA was isolated from the second sample using the NucleoSpin RNA kit (Macherey Nagel) and quantified by qPCR.

### Strand-specific cDNA synthesis and qPCR

To exclusively measure either mRNA or asRNA in qPCR, RNA-specific reverse primers were used (Supplementary Table 3) in cDNA synthesis (Nippon Genetics) and two separate samples were created. Each contained either the asRNA primer or the mRNA primer, resulting in separate asRNA and mRNA cDNAs. Furthermore, actinomycin D was added together with the reverse transcriptase because it prevents non-specific transcription from DNA and thereby secures strand-specific transcription, as reported previously[48,49]. In the qPCRs, the corresponding cDNA of mRNA and asRNA from one gene were measured with the same primer pair.

### Drop-dilution analysis

Cells were grown to the logarithmic phase (2 × 10$^7$ cells per ml) and diluted to 1 × 10$^6$ cells per ml. Then, 10-fold serial dilutions to 1 × 10$^3$ cells per ml were prepared and 8 µl of each dilution was spotted onto selective plates. The plates were incubated for 3 days at the indicated temperatures and conditions. Pictures were taken after 2 or 3 days with an Intelli Scan 1600 (Quanto Technology) and the SilverFast Ai program.

### RNA-seq

The sequencing of RNA samples was conducted at the NGS-Integrative Genomics Core Unit of the University Medical Center Göttingen.

Samples were prepared with the TruSeq RNA Sample Prep Kit v.2, according to the manufacturer's protocol (Illumina). Single-read (50 bp) sequencing was conducted using a HiSeq 4000 (Illumina). Fluorescence images were transformed to BCL files with the Illumina BaseCaller software (v.3.6.3) and samples were demultiplexed to FASTQ files with *bcl2fastq* (v.2.17).

### Differential gene-expression analysis

Sequences were aligned to the genome reference sequence of *Saccharomyces cerevisiae* (sacCer3, obtained from UCSC; https://hgdownload.cse.ucsc.edu/goldenPath/sacCer3/bigZips/) using *STAR* software[50] v.2.5, allowing for two mismatches. Subsequently, abundance measurement of reads overlapping with exons or introns was conducted with *featureCounts*[51], *subread* v.1.5.0-p1, Ensembl (EF4.68) supplemented with the coordinates of UTRs, CUTs, SUTs[22,52,53] and XUTs[3,29]. Data were processed in the R/Bioconductor environment (www.bioconductor.org, R v.3.6.1) using the *DESeq2* package[54]; v.1.24.0). The sequencing data and abundance measurement files have been submitted to the NCBI Gene Expression Omnibus database. For null-hypothesis testing, the Wald test was used with multiple comparison adjustments using the Benjamini and Hochberg method. In downstream analysis, only transcripts with an average count above 40 were considered.

### Sense–antisense-pair identification

Overlapping sense–antisense pairs were identified using BEDTools intersect (v.2.3.1)[55], requiring overlaps to occur on the opposite strand with a minimum overlap of 0.5. lncRNAs were considered in analysis as SUTs, XUTs or CUTs only if they do not overlap with other transcripts of the other types on the same strand.

### RNAi coverage analysis and classification

For gene coverage of RNAi degradation products, reads were trimmed using Cutadapt (v.2.1)[56] and aligned to the reference genome with TopHat2 (v.2.1.1)[57]. For gene coverage, the geneBody_coverage module of the RSeQC package was used (v.2.6.4)[58]. The input BED file was filtered by lncRNA classes (SUT, CUT or XUT) or by RNA enrichment in RNAi-seq. Overlapping features on the same strand were excluded. To calculate the enrichment in RNAi-seq, the read densities of a transcript in the RNAi strain was divided by its read densities in the wild type. Subsequently, the logarithm to base 2 of this ratio was calculated. For subsequent analyses, transcripts were grouped into ten groups from −5 to 5 without 0, on the basis of their $\log_2$-transformed fold change ($\log_2$ [RNAi/wild type]). Group 1 contained transcripts with changes between 0 and 1; group 2 contained changes between 1 and 2, and so on. Finally, group 5 contained transcripts that have a $\log_2$-transformed fold change above 4. In the negative range, the classification was made in the same way.

### Dimethyl sulfate reactivity analysis

The dimethyl sulfate reactivity assay was carried out as previously described[45]. The dimethyl sulfate reactivity for each transcript was summed in wild type and in *dbp2*Δ. The average reactivity in the wild type was subtracted from the average reactivity in *dbp2*Δ to obtain the structural change between the strains.

### Statistics and reproducibility

Experiments from which a significance was calculated were conducted independently at least three times. In Figs. 1h,i, 2f, 4c,h and Extended Data Figs. 4b, 7b,c,e, 8d,e,g and 9g, data are presented as mean values ±s.d., two-sided *t*-test $P < 0.05^*$, $P < 0.01^{**}$, $P < 0.001^{***}$. In Figs. 1l, 4g and Extended Data Fig. 4a, the box plots are defined by the median as the centre line, the 25th and 75th percentiles as the box boundaries and the 10th and 90th percentiles as the whiskers. Two-sided Welch's *t*-test, $P < 0.05^*$, $P < 0.01^{**}$, $P < 0.001^{***}$, $P < 0.0001^{****}$. In Figs. 1a–d, 2c,e, 3a,d, 4c and Extended Data Figs. 4c, 5a,b and 8b, the box plots are defined by

the median as the centre line, the 25th and 75th percentiles as the box boundaries and the minimum and maximum values as the whiskers. Two-sided *t*-test, $P < 0.05^*$, $P < 0.01^{**}$, $P < 0.001^{***}$, $P < 0.0001^{****}$. In Fig. 2e, the centred asterisks show a significant enrichment compared with time point 0. One-sided *t*-test, $P < 0.05^*$, $P < 0.01^{**}$, $P < 0.001^{***}$, $P < 0.0001^{****}$. Spearman's rank correlation analysis in Extended Data Fig. 5c and associated and between repetitions in Extended Data Figs. 1d, 2c and 3b were calculated using GraphPad PRISM.

### Reporting summary

Further information on research design is available in the Nature Portfolio Reporting Summary linked to this article.

## Data availability

Fractionation RNA-seq data have been deposited at the NCBI Gene Expression Omnibus (www.ncbi.nlm.nih.gov/geo/) with the accession number GSE188455. J2 RIP-seq data can be accessed with accession number GSE252951. RNA-seq of cells exposed to 0.6 M NaCl was provided previously[34] with the accession number GSE89554. RNAi-seq data have been deposited previously with the accession number GSE64090. Dbp2 iCLIP and structure-seq data have the accession number GSE106479, provided previously[45]. Source data are provided with this paper.

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

**Acknowledgements** We thank J. Beggs, R. Bordonné, P. Fabrizio, R. Ficner, R. Lill, R. Lührmann, P.A. Silver, E. Hurt, B. Seraphin, K. Weis and S. Wente for providing plasmids, strains or antibodies. This work was funded by the University of Göttingen and the Deutsche Forschungsgemeinschaft through grant 1779/12-1 and SFB 1565, Projektnummer 469281184, TP10 (to H.K.).

**Author contributions** Experiments were designed and data were interpreted by I.C. and H.K. All experiments were carried out by I.C. except: the cytoplasmic fractionation experiment for Fig. 1a–c and Extended Data Fig. 1, which was done by A.G.H.; RNA-seq and preliminary analysis by O.S. and G.S.; experiments for Fig. 2g,h were done by O.G.; and experiments for Fig. 1j–l and associated work were done by I.C and J.-P.L. S.W. conducted experiments for Figs. 2d, 4e and Extended Data Figs. 7h and 8e. The manuscript was written by H.K. All authors discussed the results and commented on the manuscript.

**Competing interests** The authors declare no competing interests.

**Additional information**
**Correspondence and requests for materials** should be addressed to Heike Krebber.

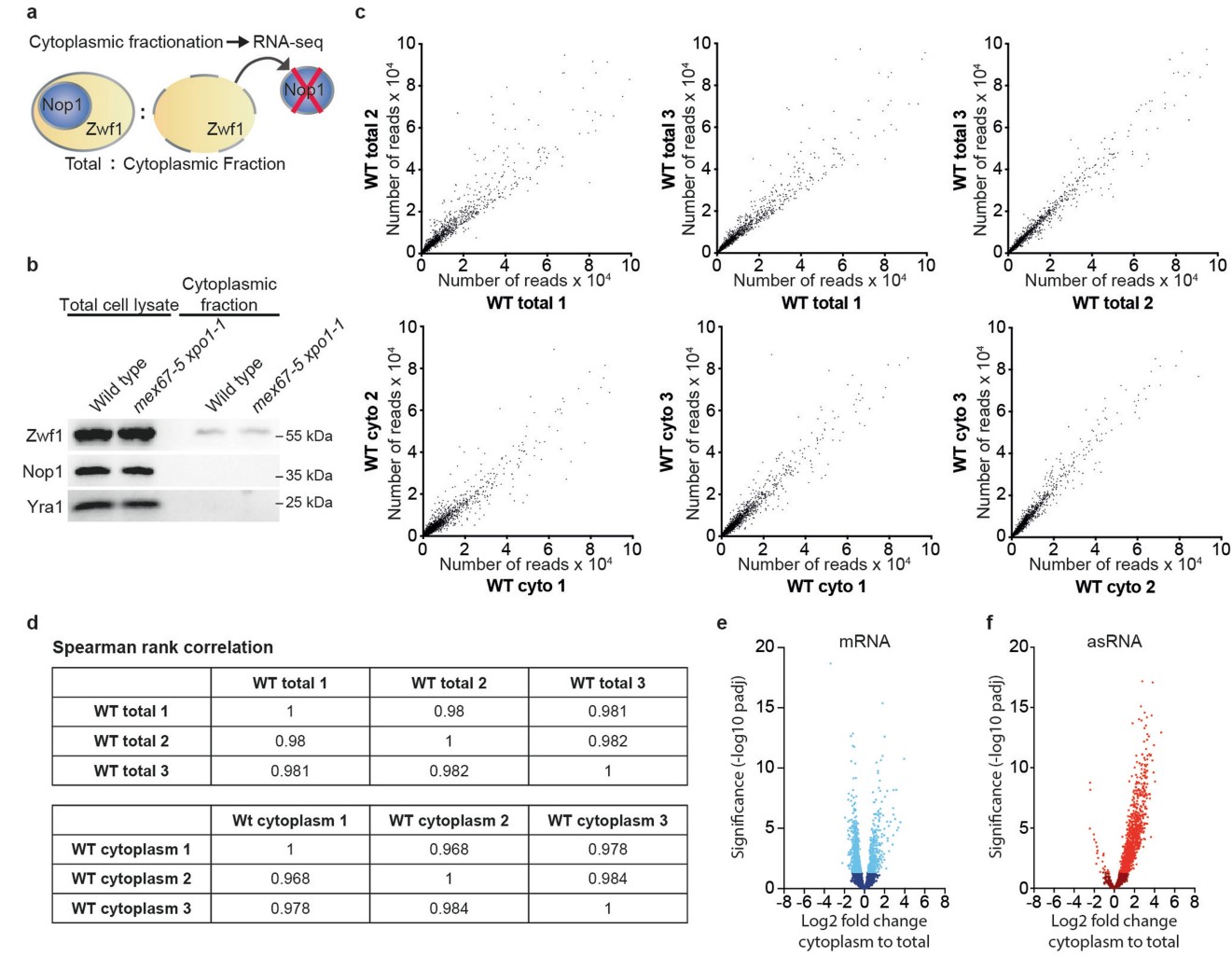

**Extended Data Fig. 1 | Supplement to Fig. 1.** (a) Cytoplasmic fractionation experiment eliminates the nuclear content of cells. RNA-seq was carried out from a total cell lysate and the cytoplasmic fraction. *n* = 3 biological independent samples. (b) Western blot analysis of nuclear and cytoplasmic proteins confirms the successful fractionation experiment. The cytosolic protein Zwf1, the nucleolar protein Nop1 and the nuclear protein Yra1 are shown before and after fractionation. (c) Fractionation-RNA-seq is highly reproducible. The read counts for each transcript across different replicates were compared against each other. (d) The calculated Spearman rank correlations between replicates are shown. (e, f) Vulcano plots of mRNAs (e) and asRNAs (f) from the fractionation-RNA-seq experiment, shown in Fig. 1a,b, are depicted. For null hypothesis testing the Wald test was used with multiple comparisons adjustments using the Benjamini and Hochberg method.

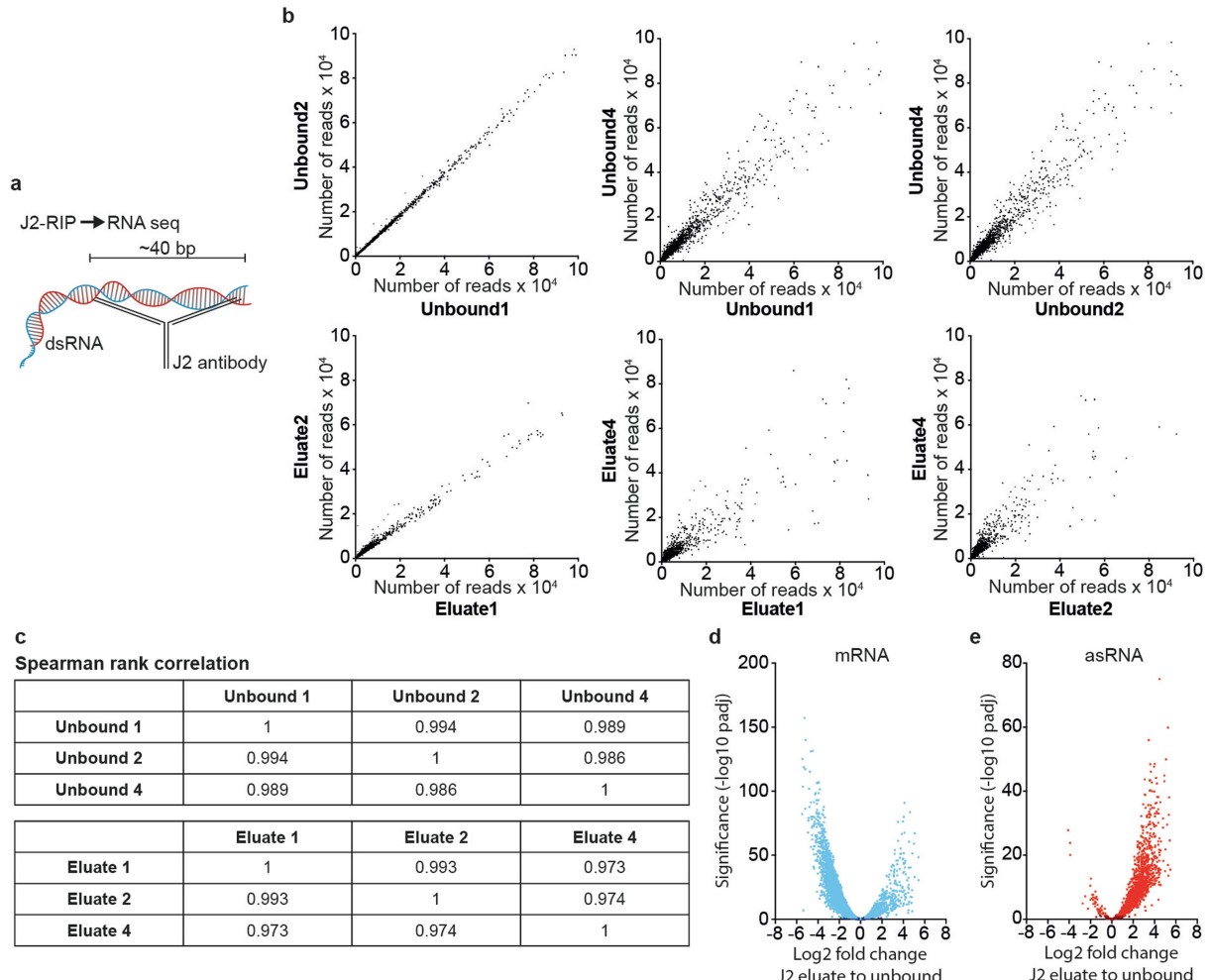

**c**

**Spearman rank correlation**

| | Unbound 1 | Unbound 2 | Unbound 4 |
|---|---|---|---|
| Unbound 1 | 1 | 0.994 | 0.989 |
| Unbound 2 | 0.994 | 1 | 0.986 |
| Unbound 4 | 0.989 | 0.986 | 1 |

| | Eluate 1 | Eluate 2 | Eluate 4 |
|---|---|---|---|
| Eluate 1 | 1 | 0.993 | 0.973 |
| Eluate 2 | 0.993 | 1 | 0.974 |
| Eluate 4 | 0.973 | 0.974 | 1 |

**Extended Data Fig. 2 | Supplement to Fig. 1.** (a) The J2 antibody recognizes ~40 bp long RNA double strands and was used to precipitate dsRNA for RNA-seq experiments. $n$ = 3 biological independent samples. (b) The J2-RNA-seq is highly reproducible. The read counts for each transcript across different replicates were compared against each other. (c) The calculated Spearman rank correlations between replicates are shown. (d, e) Vulcano plot of mRNAs (d) and asRNA (e) of the J2-RNA-seq are shown. For null hypothesis testing the Wald test was used with multiple comparisons adjustments using the Benjamini and Hochberg method.

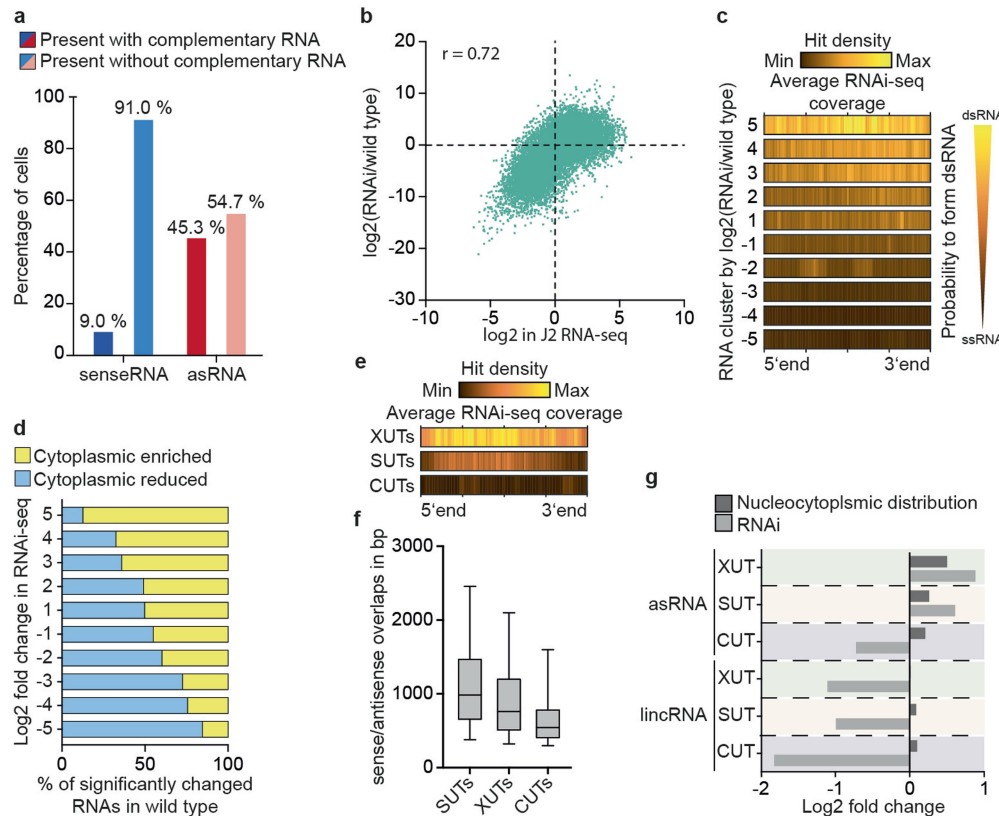

**Extended Data Fig. 3 | Supplement to Fig. 1.** (a) Most asRNAs are present in cells together with their sense transcripts. Single cell analysis of[59] was analyzed for the simultaneously presence of sense and antisense transcript of a gene in a single cell. (b) The J2-RIP-seq and the RNAi-seq results strongly correlate. The log2-fold change of the J2-RIP-seq and RNAi-seq were plotted against each other and the spearman rank correlation was calculated (r = 0.72). (c) Groups of mRNAs based on their enrichment in RNAi-seq[21] shown in Fig. 1g are depicted with their hit density of the degradation products along their gene bodies produced by the artificially expressed Dicer protein. (d) The probability of a cytoplasmic localization increases with the probability to be part of a dsRNA as determined by the RNAi-seq. Significantly cytoplasmic enriched and reduced targets of each group based on the RNAi-seq classification (Extended Data

Fig. 3c) are depicted in percent. (e) XUTs are most likely to form dsRNA. lncRNAs were grouped by their classification into SUTs, CUTs and XUTs. The degradation products of the artificially expressed Dicer were then mapped onto the gene bodies of these separate groups. (f) The length overlaps in base pairs (bp) between mRNAs and their asRNA counterparts are shown in boxplots for SUTs, XUTs and CUTs. (g) asRNAs are on average more likely enriched in RNAi-seq and to localize in the cytoplasm (by fractionation-RNA-seq) than long intergenic non coding (linc)RNAs. asRNAs and lincRNAs were defined by whether they overlap a protein coding gene on the opposite strand, classified in XUTs, SUTs and CUTs and then analyzed for their log2 fold change in the RNAi-seq and the fractionation-RNA-seq.

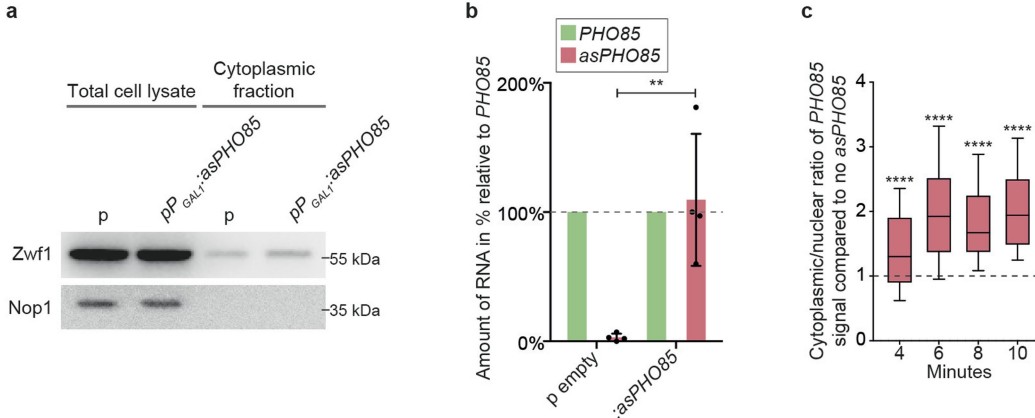

**Extended Data Fig. 4 | Supplement to Fig. 1.** (a) Western blot of the cytoplasmic fractionation experiment shown in Fig. 1i and Extended Data Fig. 4b. The cytosolic protein Zwf1 and the nucleolar protein Nop1 are shown before and after fractionation. (b) Overexpression of as*PHO85* results in similar amounts of sense and asRNA. Wild type cells either carrying an empty vector or a plasmid with *asPHO85* under the *GAL1* promoter were grown over night in glucose (uninduced) or 2% galactose (induced) to their log phase. Subsequently, the cells were lysed, the RNA isolated and quantified in qPCR. $n = 4$ independent biological experiments. (c) Quantification of smFISH at different time points after induction. The ratios of the *PHO85* mRNA signal were calculated with or without simultaneous asRNA expression and subsequently related to the mean value of cells without asRNA (dotted line). $n > 50$ cells over 3 biological independent experiments. From left to right $P = 6.79 \times 10^{-06}$, $P = 2.10 \times 10^{-13}$, $P = 4.79 \times 10^{-18}$, $P = 5.32 \times 10^{-16}$.

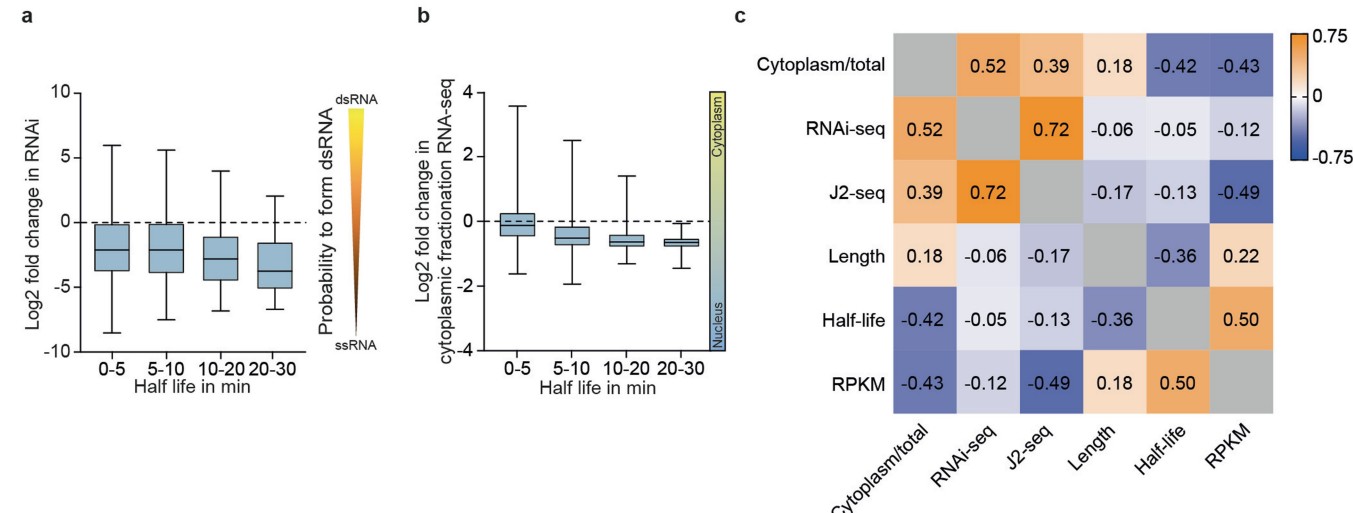

**Extended Data Fig. 5 | Supplement to Fig. 1.** (a) dsRNAs do not show increased stability. mRNAs were divided in 4 groups (from left to right n = 3320, n = 1121, n = 322, n = 50) regarding their half-life determined by Chan et al.[24]. The log2 fold change in RNAi-seq of the groups is represented. Only mRNAs that were present in both experiments were considered. The box plots are defined by the median as the center line, the 25th and 75th percentiles as the box boundaries and the Min and Max values as the whiskers. The presentation as a box plot was chosen for a clearer visualization of the respective comparison. The correlation was calculated independently of the grouping as the spearman correlation rank (r = −0.05). (b) The half-life of nuclear mRNAs is on average higher than that of cytoplasmic mRNAs. mRNAs were divided in 4 groups (from left to right n = 3489, n = 1214, n = 343, n = 53) regarding their half-life determined by Chan et al.[24]. The groups were analyzed regarding their nucleo-cytoplasmic distribution as determined in the fractionation-RNA-seq. Only mRNAs that were present in both experiments were considered. The box plots are defined by the median as the center line, the 25th and 75th percentiles as the box boundaries and the Min and Max values as the whiskers. The presentation as a box plot was chosen for a clearer visualization of the respective comparison. The correlation was calculated independently of the grouping as the spearman correlation rank (r = −0.42). (c) Heatmap of spearman correlation ranks between different data sets.

a

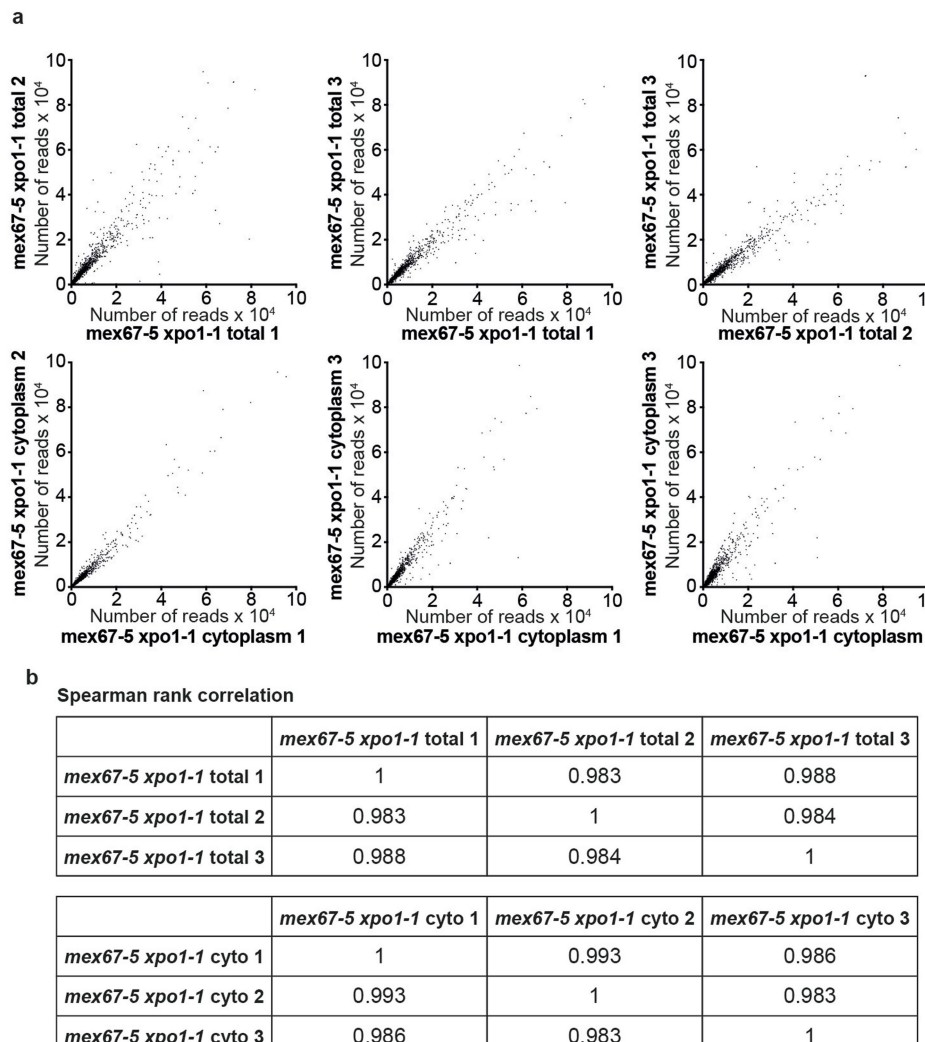

b

**Spearman rank correlation**

|  | *mex67-5 xpo1-1* total 1 | *mex67-5 xpo1-1* total 2 | *mex67-5 xpo1-1* total 3 |
|---|---|---|---|
| *mex67-5 xpo1-1* total 1 | 1 | 0.983 | 0.988 |
| *mex67-5 xpo1-1* total 2 | 0.983 | 1 | 0.984 |
| *mex67-5 xpo1-1* total 3 | 0.988 | 0.984 | 1 |

|  | *mex67-5 xpo1-1* cyto 1 | *mex67-5 xpo1-1* cyto 2 | *mex67-5 xpo1-1* cyto 3 |
|---|---|---|---|
| *mex67-5 xpo1-1* cyto 1 | 1 | 0.993 | 0.986 |
| *mex67-5 xpo1-1* cyto 2 | 0.993 | 1 | 0.983 |
| *mex67-5 xpo1-1* cyto 3 | 0.986 | 0.983 | 1 |

**Extended Data Fig. 6 | Supplement to Fig. 2.** (a) Fractionation-RNA-seq in *mex67-5 xpo1-1* is highly reproducible. The read counts for each transcript across different replicates were compared against each other. (b) The calculated Spearman rank correlations between replicates are shown.

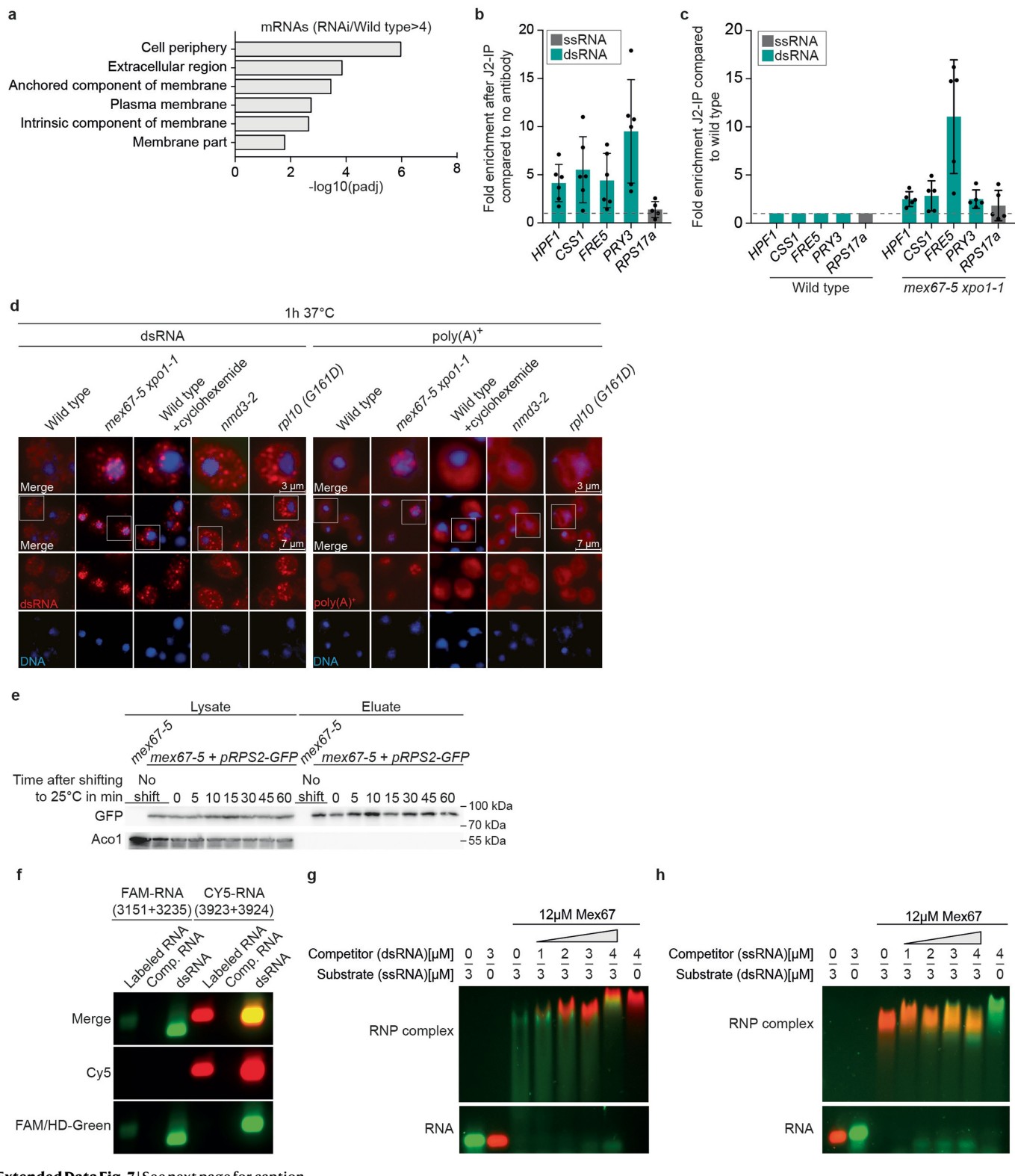

**Extended Data Fig. 7 |** See next page for caption.

**Extended Data Fig. 7 | Supplement to Fig. 2.** (a) mRNAs that belong to the top group in Fig. 2a encode for membrane proteins. GO-term analysis of this group uncovers mRNAs that possibly undergo intracellular transport. Overrepresentation was calculated with the Fishers exact test and corrected with the Bonferroni correction. (b) J2 precipitates dsRNA. J2-RIP was conducted followed by qPCR. Four highly confident dsRNA targets were chosen to verify the dsRNA pulldown by J2 in RIP experiments compared to a control with no added antibody and subsequent qPCRs. $n = 5$ biological independent experiment (c) dsRNA accumulates in the export mutant. Same experiment as shown in (b) in the indicated strains and related to J2 pulldown in wild type. $n = 4$ biological independent experiments (d) dsRNAs are exported to reach the ribosome. IF experiment with the J2 antibody (primary) and a Cy3-labelled secondary antibody and a FISH with a Cy3-labelled oligo d(T) probe are shown in the indicated strains and treatments. $n = 3$ biological independent experiments (e) dsRNA contacts ribosomes before ssRNAs. Rps3-GFP was precipitated at different timepoints before and after export release. The successful pull-down and isolation of Rps3-GFP is shown in an example Western blot. The bound RNA was purified and subsequently analyzed in Fig. 2e. $n = 3$ biological independent experiments (f) RNA substrates run differently in agarose gel electrophoresis depending on their tag and whether they are single or double stranded. RNA substrates tagged with Cy5 or FAM and their complementary untagged oligos for subsequent EMSA were loaded onto a 0.5% TAE agarose gel supplemented with HD-green. HD-green intercalates with double strand nucleic acids and thus stains the provided RNA only when a double strand was formed. In this case, HD-green staining was only visible when the tagged RNA substrate and the complementary oligo were combined, proving the formation of the double strand in vitro. $n = 1$ (g, h) Competition assay reveals a preferential binding of Mex67 to dsRNA. The EMSA shown in Fig. 2i and j were repeated with changed labels. FAM-labelled (green) ssRNA (g) or Cy5-labeled (red) dsRNA (h) were pre-incubated with low concentrations of Mex67 for complex formation. Subsequently, increasing amounts of a Cy5-labeled (red) dsRNA (left) or FAM-labeled (green) ssRNA (right) was added as a competitor. $n = 3$ independent experiments with similar results.

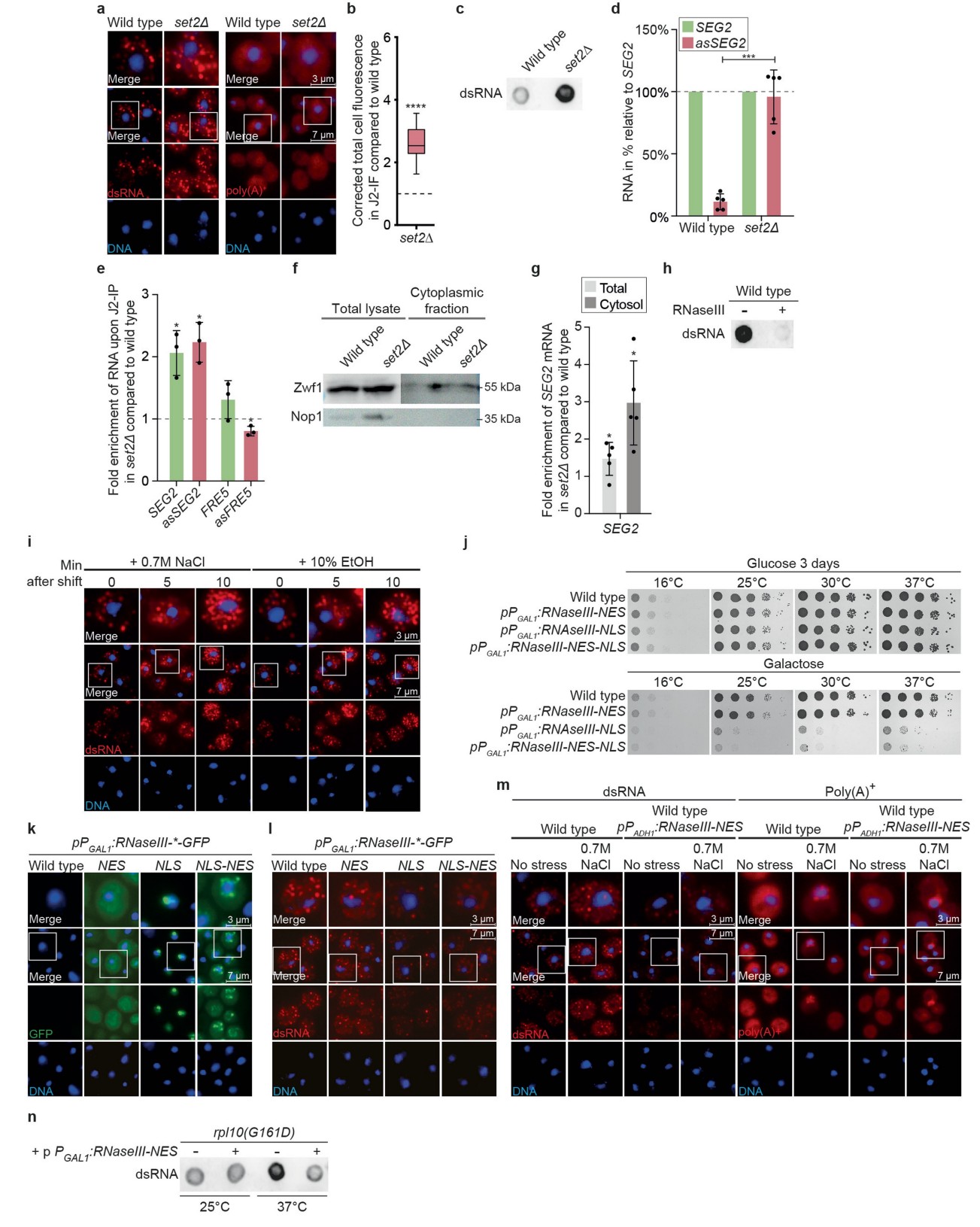

**Extended Data Fig. 8** | See next page for caption.

**Extended Data Fig. 8 | Supplement to Fig. 3.** (a) Absence of the transcription factor Set2 leads to increased dsRNA production. IF with J2-antibody and FISH with a Cy3-labelled oligo d(T) probe in indicated strains are shown. (b) Quantification of the signal intensities depicted in (a). Signal intensity was determined via Fiji. $n$ = 20 cells examined over 3 biological independent samples. P = $1.37 \times 10^{-08}$. (c) *set2Δ* contains elevated amounts of dsRNA. J2-dot blot of 1 μg RNA is shown. $n$ = 3 biological independent experiments with similar results. (d) The level of the Set2-responsive antisense RNA of *SEG2* is significantly increased in *set2Δ*. *SEG2* mRNA and asRNA amounts in wild type and *set2Δ* were measured by strand specific qPCR. $n$ = 5 biological independent experiments. (e) The *SEG2* mRNA and its asRNA form dsRNA in *set2Δ*. J2-RIPs were carried out in wild type and *set2Δ* followed by strand specific qPCR of *SEG2* and *asSEG2*. $n$ = 3 biological independent experiments. (f) Western blot of the cytoplasmic fractionation experiment shown in Extended Data Fig. 8g. (g) Increased presence of *SEG2* in the cytosol of *set2Δ* cells. Cytoplasmic fractionation experiment in wild type and *set2Δ* was carried out, followed by strand specific qPCR of indicated targets. $n$ = 5 biological independent experiments. (h) RNaseIII degrades dsRNA from yeast. Recombinant RNaseIII was added to 1 μg of total RNA isolated from wild type cells. After 20 min incubation at 37 °C the RNA was dropped onto a membrane together with a control of untreated 1 μg total RNA. Subsequently a J2 dot-blot was conducted. $n$ = 1. (i) dsRNA amounts increase under stress conditions. J2-IF with a secondary Cy3 labeled anti-mouse antibody is shown in wild type cells, exposed to the indicated stress conditions for the indicated time. $n$ = 3 biological independent experiments. (j) Directing the dsRNA degrading bacterial RNaseIII into the nucleus of yeast cells is lethal. Growth analyses of wild type cells expressing the indicated constructs on glucose (repressing) and galactose (inducing conditions) containing plates at the indicated temperatures. $n$ = 3 biological independent experiments with similar results. (k) Localization of the GFP- and transport signal-tagged RNaseIII fusion proteins in yeast cells. Plasmid containing wild type cells were grown to log phase before the RNaseIII expression was induced through the addition of 2 % galactose for 6 h. The GFP signal was detected using fluorescence microscopy. $n$ = 3 biological independent experiments. (l) J2-IF in cells expressing GFP- and transport signal-tagged RNaseIII fusion proteins. Cells were treated as in Extended Data Fig. 8k. $n$ = 3 biological independent experiments. (m) Stress-induced dsRNA is degraded by cytoplasmic RNaseIII. J2-IF and oligo d(T) FISH are shown for the indicated strains either without stress or after a 30 min incubation with 0.7 M NaCl. $n$ = 3 biological independent experiments. (n) Cytoplasmic RNaseIII-NES degrades accumulated dsRNA in the translation mutant *rpl10(G161D)*. *rpl10(G161D)* either containing an empty vector or a plasmid encoding the galactose inducible RNaseIII-NES was grown in galactose containing media at 25 °C until log phase. Half of the cells were shifted to 37 °C for 1 h before the RNA was isolated and 1 μg spotted onto a nylon membrane. The dsRNA level was detected with the J2-antibody. $n$ = 3 biological independent experiments with similar results.

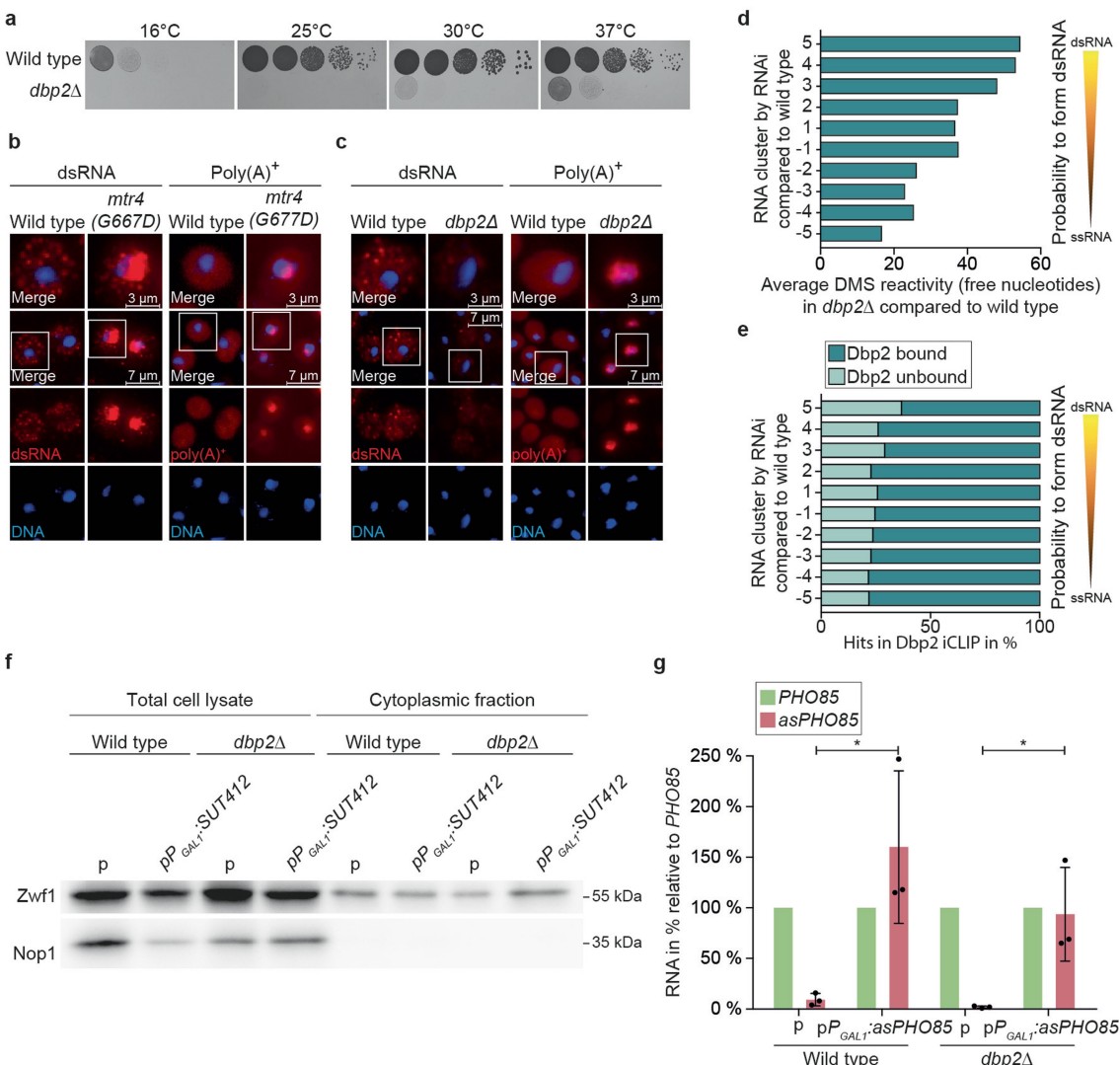

**Extended Data Fig. 9 | Supplement to Fig. 4.** (a) Deletion of *DBP2* leads to severe cold sensitivity. 10-fold serial dilutions of wild type and *dbp2Δ* after 3 days of growth on full medium plates. *n* = 3 independent biological replicates with similar results. (b,c) J2-IF and FISH with an oligo d(T) probe is shown. Strains were shifted for 2.5 h to the non-permissive temperature of *mtr4(G677D)* at 37 °C (b) or for 1 h to the non-permissive temperature of *dbp2Δ* at 25 °C (c). (d) On average, dsRNAs are less structured in *dbp2Δ*. The measured DMS reactivity in Structure-seq data from *dbp2Δ* was compared to wild type and analyzed for every group based on the RNAi-seq. *n* = 2 independent biological replicates. (e) Dbp2 binds to ssRNA and dsRNA targets in similar amounts. mRNAs with hits in Dbp2 iCLIP[45] were counted for each group based by RNAi-Seq

and are given in percentage related to the total amount of mRNAs of each group. (f) Western blot of the cytoplasmic fractionation experiment shown in Fig. 4h for the cytoplasmic protein Zwf1 and the nuclear protein Nop1 is shown. *n* = 3 biological independent experiments with similar results. (g) Overexpression of the *asPHO85* results in similar amounts of sense and asRNA in wild type and *dbp2Δ*. Wild type and *dbp2Δ* strains were either transformed with an empty vector or with a plasmid carrying *asPHO85* under the *GAL1* promoter. The strains were grown over night under inducing conditions (2% galactose) to log phase. Subsequently, cells were lysed, the RNA was isolated and used in qPCR. *n* = 3 biological independent experiments.

# Reporting Summary

## Statistics

For all statistical analyses, confirm that the following items are present in the figure legend, table legend, main text, or Methods section.

| n/a | Confirmed | |
|---|---|---|
| ☐ | ☒ | The exact sample size (*n*) for each experimental group/condition, given as a discrete number and unit of measurement |
| ☐ | ☒ | A statement on whether measurements were taken from distinct samples or whether the same sample was measured repeatedly |
| ☐ | ☒ | The statistical test(s) used AND whether they are one- or two-sided *Only common tests should be described solely by name; describe more complex techniques in the Methods section.* |
| ☒ | ☐ | A description of all covariates tested |
| ☒ | ☐ | A description of any assumptions or corrections, such as tests of normality and adjustment for multiple comparisons |
| ☐ | ☒ | A full description of the statistical parameters including central tendency (e.g. means) or other basic estimates (e.g. regression coefficient) AND variation (e.g. standard deviation) or associated estimates of uncertainty (e.g. confidence intervals) |
| ☒ | ☐ | For null hypothesis testing, the test statistic (e.g. *F*, *t*, *r*) with confidence intervals, effect sizes, degrees of freedom and *P* value noted *Give P values as exact values whenever suitable.* |
| ☒ | ☐ | For Bayesian analysis, information on the choice of priors and Markov chain Monte Carlo settings |
| ☒ | ☐ | For hierarchical and complex designs, identification of the appropriate level for tests and full reporting of outcomes |
| ☐ | ☒ | Estimates of effect sizes (e.g. Cohen's *d*, Pearson's *r*), indicating how they were calculated |

*Our web collection on statistics for biologists contains articles on many of the points above.*

## Software and code

Policy information about availability of computer code

| Data collection | Single read (50 bp) sequencing was conducted using a HiSeq 4000 (Illumina). Fluorescence images were transformed to BCL files with the Illumina BaseCaller software (version 3.6.3) and samples were demultiplexed to FASTQ files with bcl2fastq (version 2.17). |
|---|---|
| Data analysis | Sequences were aligned to the genome reference sequence of Saccharomyces cerevisiae (sacCer3, obtained from UCSC, https://hgdownload.cse.ucsc.edu/goldenPath/sacCer3/bigZips/) using the STAR software (49; version 2.5) allowing for 2 mismatches. Subsequently, abundance measurement of reads overlapping with exons or introns was conducted with featureCounts (50, subread version 1.5.0-p1, Ensembl (EF4.68) supplemented with the coordinates of UTRs, CUTs and SUTs 22,51,52 and Xrn1-sensitive unstable transcripts 3,29 Data was processed in the R/Bioconductor environment (www.bioconductor.org, R version 3.6.1) using the DESeq2 package (53; version 1.24.0). The sequencing data and abundance measurement files have been submitted to the NCBI Gene Expression Omnibus (GEO) database. For null hypothesis testing the Wald test was used with multiple comparisons adjustments using the Benjamini and Hochberg method. In downstream analysis only transcripts with an average count above 40 were considered.<br>Overlapping features respectively sense and antisense pairs were identified with BEDTools intersect (version 2.3.1) 54 requiring overlaps to occur on the opposite strand with a minimum overlap of 0.5.<br>For gene coverage of RNAi degradation products, the reads were trimmed using Cutadapt (version 2.1) 55 and aligned to the reference genome with TopHat2 (version 2.1.1) 56. For gene coverage the geneBody_coverage module of the RSeQC package was used (version 2.6.4) 57. The input BED file was filtered by lncRNA classes (SUT, CUT or XUT) or by RNA enrichment in RNAi-seq. Overlapping features on the same strand were excluded. |

For manuscripts utilizing custom algorithms or software that are central to the research but not yet described in published literature, software must be made available to editors and reviewers. We strongly encourage code deposition in a community repository (e.g. GitHub). See the Nature Portfolio guidelines for submitting code & software for further information.

## Data

Policy information about availability of data

All manuscripts must include a data availability statement. This statement should provide the following information, where applicable:

- Accession codes, unique identifiers, or web links for publicly available datasets
- A description of any restrictions on data availability
- For clinical datasets or third party data, please ensure that the statement adheres to our policy

Fractionation-RNA-Seq data have been deposited at the NCBI gene expression omnibus (GEO; www.ncbi.nlm.nih.gov/geo/) with the GEO accession number GSE188455. J2-RIP-seq data can be accessed with accession number GSE252951. RNA-seq of Cells exposed to 0.6M NaCl was provided 34 under the accession number GSE89554. RNAi-seq data has been deposited by Wery and colleagues under the accession number GSE64090. Dbp2 iCLIP and Structure-Seq data can be found under the accession number GSE106479 provided 45.

# Field-specific reporting

Please select the one below that is the best fit for your research. If you are not sure, read the appropriate sections before making your selection.

☒ Life sciences  ☐ Behavioural & social sciences  ☐ Ecological, evolutionary & environmental sciences

For a reference copy of the document with all sections, see [nature.com/documents/nr-reporting-summary-flat.pdf](http://nature.com/documents/nr-reporting-summary-flat.pdf)

# Life sciences study design

All studies must disclose on these points even when the disclosure is negative.

| | |
|---|---|
| Sample size | All statistically analysed experiments were independently repeated at least three times and determined according to standard molecular biology procedures. |
| Data exclusions | Data were excluded from analysis only due to technical failure. Western blot analysis of co-immunoprecipitation, RNA co-immunoprecipitation and cytoplasmic fractionation demonstrated the integrity of these experiments. |
| Replication | All experiments were reproducible and confirmed by statistical analysis as described. At least three independent replicates were performed for statistically analysed experiments. Samples for RNA sequencing were tested prior to sequencing. For cytoplasmic fractionation, Western blots were prepared and the distribution of known RNAs such as rRNAs and snoRNAs was tested by qPCR. For J2-RIP-seq, known dsRNAs were tested by qPCR. Validated samples were then sent for RNA sequencing. |
| Randomization | No randomization was used. Samples were prepared from a minimum of 1x10^8 cells and appropriate controls were included. |
| Blinding | Blinding was not performed. Quantification of fluorescent signal was performed on all intact cells of an image section from three independent replicates. Acquisition and quantification were performed by multiple researchers. |

# Reporting for specific materials, systems and methods

We require information from authors about some types of materials, experimental systems and methods used in many studies. Here, indicate whether each material, system or method listed is relevant to your study. If you are not sure if a list item applies to your research, read the appropriate section before selecting a response.

### Materials & experimental systems

| n/a | Involved in the study |
|---|---|
| ☐ | ☒ Antibodies |
| ☐ | ☒ Eukaryotic cell lines |
| ☒ | ☐ Palaeontology and archaeology |
| ☒ | ☐ Animals and other organisms |
| ☒ | ☐ Human research participants |
| ☒ | ☐ Clinical data |
| ☒ | ☐ Dual use research of concern |

### Methods

| n/a | Involved in the study |
|---|---|
| ☒ | ☐ ChIP-seq |
| ☒ | ☐ Flow cytometry |
| ☒ | ☐ MRI-based neuroimaging |

## Antibodies

| | |
|---|---|
| Antibodies used | anti-Nop1, Sanata Cruz, sc-57940, 28F2; anti-Myc, Santa Cruz, sc-789, A-14; anti-Yra1, Santa Cruz, yc-14; anti-Zwf1, Prof. Dr. Ulrich Mühlenhoff; anti-Hem15, Prof. Dr. Ulrich Mühlenhoff; anti-Aco1, Prof. Dr. Ulrich Mühlenhoffl; anti-Grx4, Prof. Dr. Ulrich Mühlenhoffl; anti-dsRNA, Jena Bioscience (Scicons), RNT-SCI-10010200, J2; anti-GFP, Chromotek, PABG1-100, PABG1; goat anti-mouse HRP |

conjugated, Dianova, 115-035-146; goat anti-rabbit HRP conjugated, Dianova, 111-035-144; goat anti-mouse Cy3 conjugated, Dianova, 115-165-146

| Validation | anti-dsRNA: Purity/Identity: Reducing and Non-reducing SDS-PAGE, Activity: AN-ELISA (relative activity compared to reference J2), Schönborn et al. (1991) Monoclonal antibodies to double-stranded RNA as probes of RNA structure in crude nucleic acid extracts. Nucleic Acids Res.19: 2993., Extended Data Fig. 2a, RRID:AB_2651015
anti-GFP: Purity: Affinity-purified antibody, Tested applications: Western Blot and Immunofluorescence, RRID:AB_2749857
anti-Nop1: Tested in Western blot, Source. Nop1p (28F2) is a mouse monoclonal antibody raised against a nuclear preparation of S. cerevisiae origin. (Santa Cruz)
anti-Yra1: Tested in Western blot (Santa Cruz)
anti-Myc: Tested apllication: Western blot; RRID:AB_631275
anti-Zwf1, anti-Hem15, anti-Aco1, anti-Grx4: Tested apllication: Western blot of recombinant proteins |

# Eukaryotic cell lines

Policy information about cell lines

| Cell line source(s) | HKY314: BY4741 Wild type, Euroscarf
HKY894: nmd3-2, Brune et al. 2005
HKY863; rpl10(G161D), Baierlain et al., 2013
HKY1353: mex67-5 xpo1-1, Gadal et al., 2001
HKY1399: mtr(G667D), Hackmann et al., 2014
HKY1414: pho85::kanMX4; Eurpscarf
HKY1892: PHO85-GFP; Euroscarf
HKY1898: set::kanMX4, Euroscarf
HKY2012: W303 Wild type, Drinnenberg et al., 2009
HKY2013: W303 pDCR1 pAGO1, Drinnenberg et al., 2009
HKY2065: DBP2/dbp2::kanMX4, Euroscarf
HKY2067: dbp2::kanMX4, this study |

| Authentication | Each strain was verified via growth on selective plates and by seqncing of specific genes and markers
HKY863, HKY1353 and HKY2067 were authenticated via growthtest analysis and polyA-FISH |

| Mycoplasma contamination | *Confirm that all cell lines tested negative for mycoplasma contamination OR describe the results of the testing for mycoplasma contamination OR declare that the cell lines were not tested for mycoplasma contamination.* |

| Commonly misidentified lines
(See ICLAC register) | *Name any commonly misidentified cell lines used in the study and provide a rationale for their use.* |

