## [Peer Review file · Nature]

Manuscript Title: dsRNA formation leads to preferential nuclear export and 2 gene expression.

Reviewer Comments & Author Rebuttals

Editorial Notes:

Redactions – unpublished data

Reviewer Reports on the Initial Version:

Referees' comments:

Referee #1:

Summary

The manuscript by Coban et al. describes an intriguing function for antisense transcripts in cells. Based on their analysis, they propose that antisense transcripts function in the transport of sense-coding transcripts to the cytoplasm. This model adds a new twist to the longstanding question of what the function of antisense transcripts is.

I find the data, not all, well presented, and the authors use some clever experiments to demonstrate the model. However, the manuscript mostly relies on fractionation and IF with a dsRNA antibody, which makes some the data heavily reliant on these two techniques. In my view, the manuscript could benefit from an additional evidence using, for example, RNA-FISH for specific transcripts instead of RT-qPCR (not sure if this is feasible on with dsRNA). An additional general comment is that sometimes additional controls could be included; also microscopy quantification is lacking mostly, and experimental details sometimes lacking. I have tried to indicate some of these below.

Fig. 1: Regarding the fractionation, while the nucleolar marker will determine whether the nucleolus is fractionated away, it would be good to have a nuclear marker. E.g. any histone protein as for evidence that the fractionation works. Fig. 1b: The effects are modest. E.g. rRNA shows 2-fold enrichment, while the other transcripts show almost no effect or an effect with a large spread. Which differences are significant? Having said that, it is challenging to perform these types of experiments.

Regarding the fractionation RNA-seq, it was not clear how many biological repeats were done. How many transcripts were significantly different, and how reproducible was the analysis?

The authors bring up the example of XUTs in the abstract and introduction, but as far as I know they

are not the only source of antisense transcripts. The authors think this mechanism is specific to XUTs; what about SUTs and CUTs, and other antisense transcripts? If there is no clear reason the exclude these, the authors could consider to rewrite these sections.

The current model is that antisense transcription tunes sense transcription, but this is not mentioned in the current manuscript perhaps. The authors can refer to this literature in the intro or discussion.

It is hard to see from Fig. S1c what the correlation is. Perhaps it is better to do the reverse analysis and split the data into groups with boxplots to make the correlation more apparent.

In Fig. 1f,g, Pho85 antisense is being expressed, which results to more cytoplasmic sense. Can the authors confirm the existence of dsRNA of Pho85? Is it possible to treat extracted RNA with RNase III in vitro? Or use J1 antibody?

Similarly, for Fig. S2a, to show the antibody recognizes double-stranded can isolated RNA be treated with RNase III? Is this a well established reagent?

Fig. 2a: While great to show example cells, quantification of certain number of cells is needed. Does treating cells with cycloheximide show the same effect? This is a general comment for the microscopy experiments.

Fig. 2b: Why does the signal drop at 30 min in WT?

Regarding the in vitro assay and Fig. 2c-e: I could not find information on the sequences used, and how it was assessed that the double SS RNA is indeed what it is.

Also in Fig. 2d,e it is formally possible that labelling of FAM and Cy3 has an effect. The label of 2e is missing.

In Fig. 3a,b the microscopy is quantified, but not in Fig. 3f,i. Can the authors comment why the J1 antibody forms speckles like shapes in cells?

Regarding Fig. 3d: This relates to the main comment. Is it possible to perform RNA-FISH? This would make it more convincing that transcripts are retained in the nucleus.

Fig. 3h needs quantification. I am not so convinced the NES is functional as is stated in the text based on the example image.

Regarding Fig. 4a, if I understand correctly the IP is dsRNA-dependent. Is the interaction indeed lost when the sample is treated with RNase?

Again quantification of microscopy is appropriate for Fig. 4b.

Referee #2:

The paper by Coban et al. presents interesting novel evidence that the formation of double-stranded RNA (dsRNA) by hybridization of mRNA with an antisense RNA may increase mRNA export efficiency.

In Fig. 1, the authors use nucleo-cytoplasmic fractionation to show that mRNAs described to form dsRNAs accumulate more efficiently in the cytoplasm, and their export is affected by *mex67-5* and *xpo1-1* export receptor mutations. Interestingly, over-expression of a PHO85 antisense RNA (asRNA) from a plasmid promotes the PHO85 mRNA cytoplasmic localization.

In Fig. 2, the authors use immunofluorescence with the J2 antibody specific for dsRNA to show that cytoplasmic dsRNA increases in *nmd3Δ* and *rpl10* mutants, suggesting that cytoplasmic dsRNA levels are modulated at the translational level. Another experiment uses RIP to examine the binding of ssRNA or dsRNA to ribosomal protein Rps2 after release of the temperature sensitive *mex67-5 xpo-1* mRNA export mutant from 37 °C to 25 °C. The data indicate that dsRNAs become bound to Rps2 faster than ssRNAs, suggesting preferential export or recognition by ribosomes. Moreover, well-designed EMSA competition experiments provide evidence that the purified Mex67-Mtr2 heterodimeric export receptor preferentially binds dsRNA.

Fig. 3 examines dsRNA formation in cells submitted to stress. Using J2 immunofluorescence, they show cytoplasmic accumulation of dsRNA in *set2Δ*, a mutant promoting asRNA transcription. Nucleo-cytoplasmic fractionations combined with RNA-seq experiments further indicate that *set2Δ* promotes asRNA production and stimulates export of sense mRNAs. Similar phenotypes are observed following salt or ethanol stress. In Fig. 3, the authors manipulate the localization of RNase III and provide further evidence that stress leads to formation and export of dsRNA in the cytoplasm that is important for cell survival.

Finally, Fig. 4 provides strong evidence that loss of the helicase Dbp2 interferes with dsRNA formation and the enhanced export of PHO85 mRNA upon PHO85 asRNA induction.

This paper uncovers very interesting new ideas and potential mechanisms; however, some experiments are poorly described and not always well controlled. The conclusions drawn from the observations are sometimes too strong. For this work to be considered for publication in Nature, additional experiments are required to validate the proposed model.

Major comments:

1. Many conclusions are based on immunofluorescence experiments using the J2 antibody described to recognize dsRNA of >40 bp. However, no control experiment in the paper really shows what type of dsRNAs are recognized by this antibody; does the signal mostly correspond to mRNA-asRNA hybrids or to other types of dsRNAs? There is no mention of how many mRNAs may form dsRNAs due to antisense (as) transcription under normal or stress conditions. More generally, how long should the mRNA-asRNA hybrids be to be efficiently exported?

2. There should be additional experiments showing the presence of both sense mRNAs and

antisense transcripts in the same cells by single-molecule FISH to get an idea of the prevalence of this mechanism under normal versus stress conditions.

a) More specifically, Fig. 1 should include smFISH experiments of PHO85 sense and antisense RNAs in the presence of p empty or p PGAL1:SUT412.

b) Fig. 3 should also include smFISH experiments revealing specific sense and antisense RNAs (i.e. SEG2 and others) in wt, set2 Δ , and specific stress conditions.

c) Finally, Fig. 4 should include smFISH experiments of PHO85 sense and antisense RNAs in wt and dpb2 Δ in the presence of empty plasmid or PGAL1:SUT412. It would also be very interesting to perform the same experiment following Dbp2 over-expression. If the model is correct, one should be able to observe an even greater increase in PHO85 mRNA export compared to wt when PHO85 asRNA is induced.

Minor corrections and questions to be addressed:

Line 58: these cytoplasmic lncRNAs

Line 63: corresponding (regarding) mRNAs

Fig. 1c: How many different sense and antisense transcripts are considered in this analysis; is the C/N ratio based on the same sequencing as in Fig. 1b?

Based on these sequencing data, how should one evaluate the number of cells producing just sense, AS or both?

Line 78: How many AS RNAs are more abundant than the corresponding mRNA, which is more cytoplasmic than an mRNA having lower levels of AS.

How many different sense-AS pairs are these analyses based on. How were these analyses done?

Fig. 1d: What sequencing data were used? How many dsRNAs (s/as pairs) were in each of the 10 groups? This information is not mentioned in the Materials and Methods.

Fig. 1d, "the more an RNA is prone to form a double strand the more it is localized to the cytoplasm in wild type cells": Is this true for both mRNAs and asRNAs?

Fig. 1e, "all groups of RNAs, including the dsRNAs were shifted to the nucleus": Are all RNAs affected the same? Based on the histogram, it seems that the RNAs with the highest probability to form dsRNA are less affected by mex67-5 xpo1-1; is that right?

Fig. 1g: y-axis should say PHO85 mRNA instead of PHO5 mRNA (same in Fig. 4d).

There is the same amount of unspliced in total cell and cytoplasm when overexpressing PHO85

asRNA, which is unexpected. One would expect more unspliced in the total vs cytoplasmic fraction, since unspliced should be retained in the nucleus. How does it look in the presence of p empty, when PHO85 asRNA is not over-expressed?

Ext. Data Fig. 2: The quantification of dsRNA vs ssRNA strand-specific cDNA synthesis and qPCR has to be explained in more detail.

Fig. 2a: As already raised above, what is known about the nature and length of dsRNAs revealed with the J2 antibody? What fraction represents mRNAs compared to other types of transcripts?

The dsRNAs seem to accumulate within cytoplasmic foci, which become more frequent and intense in the *nmd3Δ* and *rpl10* mutant backgrounds (and even more so in *set2Δ*, Fig. 3a). What do these aggregates/condensates correspond to?

Why is there also more polyA cytoplasmic signal in *nmd3Δ* and *rpl10* mutants? Are the dsRNAs more stable? Are the dsRNAs adenylated?

Fig. 2b: Based on what criteria have the ss- and dsRNAs analyzed in this experiment been chosen? Why are there seemingly two waves of dsRNA binding to Rps2 following the shift from 37 °C to 25 °C? It increases until 10 min followed by a decrease, but it goes up again at 60 min. The standard deviations of these measurements are quite big depending on the time point. Why is that?

Fig. 2c: The EMSA experiment is not well described, neither in the text nor in Materials and Methods. What length are the ss- and dsRNAs used in this experiment? How many Mex67-Mtr2 heterodimer molecules can bind the ss- or dsRNA? What are the absolute molarities used in this experiment?

It would also be useful to put a size marker to indicate how the ssRNA and dsRNA EMSA assays compare.

Fig. 2d: p. 8 line 189, It should say Fig. 2d and not Fig. 2e (there is no Fig. 2e).

Fig. 3e: p. 9 line 214 “the mean of the sense RNAs rather stayed the same and showed specific sense transcript upregulation”: How many sense and asRNAs were considered in this analysis? How many genes show a substantial increase in antisense transcription upon shift to 0.6 M NaCl? Which specific sense transcripts were upregulated, and was their upregulation also enhanced in *set2Δ*?

Fig. 3h: The dsRNA signal in the wild type appears much stronger than in other wild-type controls such as Fig. 2a, 3a, or 3f.

Fig. 4a: Describe in the legend whether these cells were grown in galactose to induce MYC-DBP2 before J2 CoIP.

p. 12: There is a wording problem: “Clearly, in the absence of Dbp2 the previous effect that overexpression of the asRNA shifted the localization of the sense RNA to the cytoplasm (Fig. 4c, 4d)” was abrogated?

Referee #3:

Krebber and co-workers describe a novel and intriguing role for asRNA in yeast: facilitation of nuclear export of their sense mRNA counterparts. The authors perform transcriptomic and traditional molecular biological experiments and find that dsRNA formation between asRNA and their complementary mRNA counterparts promotes export of these mRNAs. They further implicate the RNA helicase Dbp2 in the formation of the dsRNA.

The manuscript reports a foundational discovery with potentially far-reaching implications for our understanding of how gene expression is regulated at the RNA level. The data also provide a novel, largely unanticipated explanation for the role of asRNAs, a longstanding question in biology. The experiments are well rationalized, and approaches and data are well explained.

Strengths of the study include several well-conceived and executed orthogonal approaches demonstrating the claimed link between asRNA-mRNA binding and promotion of mRNA export. Weaknesses of the study are potential problems in the assignment of a role for Dbp2 in the formation of the dsRNA, and the heavy reliance on the J2 antibody for the detection of dsRNA in cells, as outlined in more detail below. It seems advisable to address these and some more minor issues (also outlined below) prior to publication. Assuming that the weaknesses are addressed, the manuscript is likely to appeal to the broad readership of Nature.

Comments / major issues:

1) The final major point of the manuscript is the suggestion that the RNA helicase Dbp2 promotes formation of the dsRNA between the mRNAs and their corresponding asRNAs. Based on the data presented, the claim that Dbp2 plays a direct role in the dsRNA formation appears premature, for the following reasons:

a) The first key experiment is a co-IP of Dbp2p with the dsRNA binding J2 antibody (Fig. 4a), based upon which the authors claim evidence for binding of Dbp2 to dsRNA in vivo (lines 261-262). This claim is an overly optimistic interpretation of the data. Dbp2 can bind to any RNA (structured or unstructured) associated with a dsRNA region, to other proteins associated with any such RNA region, or a combination thereof. Given that alternative scenarios are consistent with the observed data, direct evidence is needed to conclusively demonstrate that Dbp2 binds to dsRNA in vivo. Since Dbp2 is a DEAD-box helicase that at least transiently binds dsRNA in order to unwind duplexes, some level of dsRNA binding is inherently expected. The point here is that the data shown are insufficient to support dsRNA binding by Dbp2 in vivo.

b) The authors then use a *dbp2* deletion strain to examine changes in the accumulation of dsRNA in the cytoplasm. Here again, the data are consistent with the claim that Dbp2 is a key factor for dsRNA formation and preferential export (lines 272-273). A direct role of Dbp2 in dsRNA formation is clearly implied by the authors, but Dbp2 might work well upstream of the actual dsRNA formation. Dbp2 is well known to act on a multitude of RNAs, and its deletion likely affects many processes, which might include production of one or more factors directly involved in dsRNA formation. More specific evidence for a direct role of Dbp2 in dsRNA formation appears necessary, especially given that Dbp2

has been implicated in the unwinding of asRNA-mRNA complexes (PMID: 26805575), and because of the pleiotropic effects of a *dbp2* deletion. The use of glucose deprivation, which moves Dbp2 to the cytoplasm, might alleviate some confounding issues. In addition, there are several CLIP datasets for Dbp2-RNA binding in yeast. Examination of whether and how the Dbp2 binding sites in cells correspond to mRNA-asRNA binding regions might be instructive for assigning a physical function of the protein in dsRNA formation.

2) The authors rely heavily on the J2 antibody to detect dsRNA in cells. The relevant microscopy data are very qualitative, and it is not immediately apparent how robust the microscopy data are. In some cases, the microscopy experiments are validated by specific, gene-focused experiments. In other cases, the microscopy data stand alone. As the localization of dsRNA is a central point in the ms, the microscopy data should be quantified, and robustly validated in all cases by an orthogonal approach.

Comments/minor issues (no particular order):

3) Line 94: The authors indicate that mRNAs with long half-lives accumulate in the nucleus, implying that these mRNAs are not preferentially translated. This observation seems to contradict observations by other groups showing increases in half-lives with preferential translation of mRNAs (e.g. PMID: 25768907). Clarification on this issue is needed.

4) Issues with data presentation in figures.

a) Fig. 1b: How were the dsRNA and ssRNA quantified? Is the difference in cytoplasmic vs. nuclear distribution statistically significant, and if so, how was this determined?

b) Fig. 1c: Is the difference in cytoplasmic vs. nuclear distribution statistically significant, and if so, how was this determined?

c) Fig. 1d: The y-axis notes probability, which is by definition a value between 0 and 1, but negative values are given. Clarification is needed.

d) Fig. 1e: Is the change in nuclear accumulation statistically significant, and if so, how was this determined? As above, the y-axis notes probability, which is by definition a value between 0 and 1, but negative values are given. Clarification is needed.

5) The gel-shift experiments (Fig. 2c,d) are of good visual quality but lack sufficient rigor.

a) The authors should report the reproducibility of the data and

b) try to quantify the affinities (Fig. 2c), in order to allow assessment of differences in binding efficiency, rather than qualitatively describe the gel shifts.

Author Rebuttals to Initial Comments:

Point by point response to the reviewers' comments on the Nature submission:

We are grateful for getting the chance for this revision and we are thankful for the valuable comments of the editor and the reviewers that we have fully addressed (see below). Please see our specific responses (in black) to the editors and reviewer's comments (in green) below. The page and line numbers are taken from the final manuscript (in blue).

All comments were very helpful and we believe that the added new work has significantly improved the quality of this manuscript. Every aspect of this work was validated and verified through various experimental procedures. In response to the reviewers' inquiries, we have incorporated new data as outlined below (see responses to the reviewers' comments). Additionally, we have included further novel figures that we considered as necessary and valuable for the paper. As we suggest an asRNA-mediated boosting mechanism for gene expression, we show the elevated protein level now in a new western blot (see new Fig. 2f). Moreover, we included studies with the helicase Mtr4 to exclude it as the dsRNA forming helicase in cells (see new Fig. 4b-d) and highlight its different function compared to Dbp2. Finally, we analyzed supportive transcriptome-wide bioinformatical studies to confirm Dbp2 as the dsRNA forming factor (see new Extended Fig. 8e).

The main reason why this revision needed more time than expected, is because the smFISH experiments were very challenging. As outlined in detail below, the problem was intrinsic, as dsRNAs significantly impede the binding of a ssDNA probe.

[REDACTED]

However, as explained in detail below, we found a solution and were able to present the requested smFISH experiment. Most importantly, it visualizes the antisense mediated boosting of gene expression.

We continue to believe that we have made a significant discovery and that our findings represent a major advance in science and we hope that the reviewers agree with a publication in nature.

Referees' comments:

Referee # 1:

Summary

The manuscript by Coban et al. describes an intriguing function for antisense transcripts in cells. Based on their analysis, they propose that antisense transcripts function in the transport of sense-coding transcripts to the cytoplasm. This model adds a new twist to the longstanding question of what the function of antisense transcripts is.

I find the data, not all, well presented, and the authors use some clever experiments to demonstrate the model. However, the manuscript mostly relies on fractionation and IF with a dsRNA antibody, which makes some the data heavily reliant on these two techniques. In my view, the manuscript could benefit from an additional evidence using, for example, RNA-FISH for specific transcripts instead of RT-qPCR (not sure if this is feasible on with dsRNA). An additional general comment is that sometimes additional controls could be included; also microscopy quantification is lacking mostly, and experimental details sometimes lacking. I have tried to indicate some of these below.

We thank this reviewer for acknowledging the importance of our findings. We agree with the criticism that the J2 antibody was often used and characterization of its targets is missing. Therefore, we added an RNA-seq experiment, in which we analyzed the targets of the antibody. As outlined below in more detail, the identified RNAs in J2-RNA-seq showed a high correlation with the RNAi-seq identified RNAs (see new Extended Fig. 3b).

Establishing the smFISH experiments was quite challenging. To visualize the RNA targets in cells, between 12 and 15 probes were needed. Since the probes seem to bind only with

difficulty in the dsRNA region, we had to design them to hybridize at the overhangs. However, these overhangs are usually not long enough to place 12 to 15 probes, which undergo also quenching, when they are too close to each other. Therefore, we had to artificially extend the length of the overhangs.

[REDACTED]

Despite these challenges, we ultimately succeeded in creating constructs that made informative smFISH analysis possible. (see below).

Moreover, we included additional controls and quantifications as requested.

Fig. 1: Regarding the fractionation, while the nucleolar marker will determine whether the nucleolus is fractionated away, it would be good to have a nuclear marker. E.g. any histone protein as for evidence that the fractionation works.

We added Yra1, a well-studied nuclear RNA binding protein, as a nuclear control for the cytoplasmic fractionation experiment. See new Extended Figure 1a.

Fig. 1b: The effects are modest. E.g. rRNA shows 2-fold enrichment, while the other transcripts show almost no effect or an effect with a large spread. Which differences are significant? Having said that, it is challenging to perform these types of experiments.

We have incorporated volcano plots for mRNAs and asRNAs to visually depict the distribution and significance of these targets in the RNA-seq data from cytoplasmic fractionation (see new Extended Figure 1d and 1e). Additionally, we have introduced analyses that specifically focus on targets with significant changes (see new Figure 1c and Extended Figure 3d).

The wide range of changes in the nucleo-cytoplasmic distribution within the mRNA and asRNA groups is indeed remarkable, for instance compared to snoRNA und rRNA. We also noticed this and therefore asked ourselves what influences the distribution of mRNAs and asRNAs. For snorNAs and rRNAs, it is of course mainly the site of place of function. The distribution of mRNAs and asRNAs, that are transcribed in the nucleus, exported to the cytoplasm and translated and degraded at the ribosome, could be influenced by several factors. On the one hand, localization is influenced by the preferential export described in this article. dsRNAs are exported faster than ssRNAs and thus, dsRNAs are more cytoplasmic. Additionally, we found that the expression level plays a role. The transcripts of highly expressed genes tend to be nuclear. This is probably due to the already described buffer mechanism of nuclear export, which compensates for the burst-like character of transcription. Hereby, the number of transcripts arriving in the cytoplasm gets rather uniform to ensure an even amount of protein (Bahar Halpern et al., 2015). Another effect on the nucleo-cytoplasmic ratio of RNAs is the further transport within the cytoplasm. mRNAs of membrane-bound proteins stay longer in the cytoplasm which shifts their nucleo-cytoplasmic ratio towards the cytoplasm compared to the mean value. Thus, various factors influence the nucleocytoplasmic distribution of mRNAs and asRNAs, resulting in a large spread in this class of transcripts. We have seen great potential in the difference between dsRNA and ssRNA and describe this mechanism in more detail here. Cytoplasmic transcripts are significantly more likely to be double-stranded transcripts and thus we got the idea that being part of a dsRNA could make a difference in the export efficiency.

The relatively "modest" effects that we observe are a result of comparing RNA in a total cell lysate to the RNA in the cytoplasmic fraction, which represents a portion of the total content. We do not compare two completely different samples, but compare the total amount of RNA with a fraction of it, which weakens the difference in total. We chose this method over the isolation of yeast nuclei because the method is more robust and reliable.

Regarding the fractionation RNA-seq, it was not clear how many biological repeats were done. How many transcripts were significantly different, and how reproducible was the analysis?

Cell fractionation was carried out with three biological replicates and to demonstrate the reproducibility of the cell fractionation and the newly conducted J2-RNA-seq, we have calculated Spearman correlation ranks between replicates (see new Extended Fig. 1a, 1b, 2a, 2b and Extended Table 1 and 2).

As described above, we added volcano-plots for mRNAs and asRNAs in extended Figure 1d, 1e, 2c and 2d to highlight the distribution of significant targets of both groups and their number and added analysis regarding only significant changed targets. In addition, we also mention absolute numbers and percentages of significant changed targets in the text.

The authors bring up the example of XUTs in the abstract and introduction, but as far as I know they are not the only source of antisense transcripts. The authors think this mechanism is specific to XUTs; what about SUTs and CUTs, and other antisense transcripts? If there is no clear reason to exclude these, the authors could consider to rewrite these sections.

This is a good point as this information was hidden before. The boxplot of the group of “asRNA” shown in Fig. 1b consists in fact of all kinds of asRNA transcripts including the SUTs, CUTs and XUTs. For more clarity, we added a graph showing the mean RNAi and cytoplasmic enrichment for CUTs, SUTs and XUTs separately (see new Extended Fig. 1j and k). From this analysis it becomes evident, that especially XUTs are most likely part of dsRNAs and enriched in the cytoplasm. We rewrote the section for more clarity.

104 “Gene coverage analysis of the RNAi-seq data furthermore revealed that these dsRNAs are mostly formed with XUTs, which show the highest cytoplasmic enrichment within the lncRNAs, fewer with transcripts identified as stable unannotated (SUTs) and even less with cryptic unstable transcripts (CUTs).”

The current model is that antisense transcription tunes sense transcription, but this is not mentioned in the current manuscript perhaps. The authors can refer to this literature in the intro or discussion.

We rephrased the text accordingly and give additionally literature examples.

49 “Several reports suggest nuclear functions for lncRNAs in *Saccharomyces cerevisiae* in regulating coding genes by suppressing transcriptional leakage⁶⁻⁸ or by inducing transcription in response to environmental changes^{9,10}.”

It is hard to see from Fig. S1c what the correlation is. Perhaps it is better to do the reverse analysis and split the data into groups with boxplots to make the correlation more apparent.

This was an excellent suggestion, which we gladly adopted. To make the trend in Fig. S1c more noticeable, we created groups of RNAs based on their half-life and show their nucleocytoplasmic distribution as boxplots (see new Extended Figure 5b).

In Fig. 1f,g, Pho85 antisense is being expressed, which results to more cytoplasmic sense. Can the authors confirm the existence of dsRNA of Pho85? Is it possible to treat extracted RNA with RNase III in vitro? Or use J2 antibody?

To address this point and to add another method that supports the possible existence of dsRNA, we conducted MS2-loop IPs that show the dsRNA formation between the *PHO85* mRNA and its antisense RNA, *asPHO85*. For this, *PHO85* was tagged with MS2-loops and expressed in cells that also express a GFP tagged MS2-coating protein. By RIP of the MS2-

protein, we precipitated the *PHO85* mRNA and co-precipitated the *asPHO85* (see new Fig. 1j).

Similarly, for Fig. S2a, to show the antibody recognizes double-stranded can isolated RNA be treated with RNase III? Is this a well-established reagent?

RNase III is a well-established agent for degrading dsRNA, and it has been shown to degrade dsRNA, recognized by the J2 antibody in *E. coli* (Lybecker et al., 2014). In their study, Lybecker and colleagues isolated dsRNA using the J2 antibody from both wild type and an RNaseIII mutant *E. coli* strain and discovered that the dsRNA increased tenfold in the RNaseIII mutant strain, supporting the dsRNA degradation capability of this enzyme.

To confirm that RNaseIII can also degrade dsRNA of *S. cerevisiae*, we conducted J2 dot blot experiments with and without the addition of recombinant RNaseIII. As the signal was significantly reduced with the addition of RNaseIII (see Extended Fig. 7h) we conclude that the enzyme is able to degrade dsRNA in general. Also *in vivo* expression of RNaseIII degraded dsRNA in *S. cerevisiae* (Fig. 3f, Extended Fig. 7l).

Fig. 2a: While great to show example cells, quantification of certain number of cells is needed. Does treating cells with cycloheximide show the same effect? This is a general comment for the microscopy experiments.

We added the requested quantifications to the J2-immunofluorescence in the new Fig. 2c. Moreover, we added a new experiment, in which we blocked translation by the addition of cycloheximide and found a similar increase of dsRNAs in the cytoplasm of treated cells. These findings support the model in which translation is required to detach dsRNAs (Fig. 2b-d).

Fig. 2b: Why does the signal drop at 30 min in WT?

The export of RNAs is mediated by Mex67 that is recruited by multiple RNA-binding proteins (RBPs) including the guard proteins, upon quality control. They are either released from the RNA directly after export or during translation. Subsequently, they are reimported into the nucleus for another round of transport. This circuit maintains a sensitive balance between the cytoplasmic and nuclear pools of the RBPs and Mex67. We hypothesize that when export is blocked and subsequent released, an unusual large quantity of RBPs is exported simultaneously, temporarily disrupting the equilibrium between the nuclear and cytoplasmic RBP content. After the initial export round, the cytoplasmic RBPs must undergo recycling and re-import, a process that takes time and leaves newly transcribed nuclear RNAs unattended. This might result in the decline observed at 30 minutes. After 60 minutes, the equilibrium between nuclear and cytoplasmic RBP-content is re-established and returns to its initial state.

Regarding the in vitro assay and Fig. 2c-e: I could not find information on the sequences used, and how it was assessed that the double SS RNA is indeed what it is.

They were listed in Table 1, however, not particularly marked. We have now created a separate table for oligos used in the EMSA (Table 6). Additionally, we have added the length of the dsRNA in the methods section. Validation of the dsRNA formation is based on its altered electrophoretic mobility in native gels as it carries a higher negative charge than the ssRNA leading to a faster movement compared to ssRNA. Furthermore, dsRNA can be visualized through staining with intercalating reagents such as ethidiumbromide or HDgreen, which we used (see Extended Fig. 6f).

Also in Fig. 2d,e it is formally possible that labelling of FAM and Cy3 has an effect. The label of 2e is missing.

To exclude such effect, we did the reverse experiment. We added an EMSA in which the FAM label was placed on the ssRNA and the Cy5 label on the dsRNA with the same result (see new Extended Figure 2e and f). Moreover, we labelled each figure separately to avoid confusion.

In Fig. 3a,b the microscopy is quantified, but not in Fig. 3f,i. Can the authors comment why the J1 antibody forms speckles like shapes in cells?

As requested, we added quantifications to all microscopy experiments (see new Fig. 1n, 2c, 3d, 4c, 4g and Extended Figure 7b). The speckle-like staining of the J2 antibody in the cytoplasm is most likely due to dsRNAs that await translation. Such “J2”-bodies were to our knowledge not described before. We investigated whether they are overlapping with P-bodies or stress granules by co-localization of the regarding marker proteins Edc1 and Pub1, respectively, but could not detect any overlap. Due to the space limitations, we did not add this negative result to the manuscript, but can certainly do this upon request.

As the J2 antibody recognizes dsRNA sequences of around 40 nt length, several antibodies bind to one dsRNA. On a dsRNA of for example 1200 bp, 30 antibodies can bind. The Cy3-coupled secondary antibody will thus concentrate the signal on this restricted area which might lead to the speckle like staining.

Regarding Fig. 3d: This relates to the main comment. Is it possible to perform RNA-FISH? This would make it more convincing that transcripts are retained in the nucleus.

We thought that this was a very good idea and we started experiments right away, but as we learned over the time the smFISH with dsRNAs was a highly challenging experiment that took us quite some time to establish, yet yielded in numerous important insights. Generally, for

successful usage of this technique, probes must hybridize with the RNA. However, when complementary RNA strands (sense and antisense) are already hybridized to each other, the probe does not bind. As already identified by Wery and colleagues (2016) through the RNAi-seq data and confirmed by our J2-seq analysis, double stranded RNAs that are formed in yeast are predominantly formed over the entire length of both, the sense and asRNA. This leaves only minimal overhangs on the pairs for possible probe annealing, as these overhangs are usually not longer than ~100 nucleotides. To visualize a single RNA in yeast a set of at least 12 probes are required, each labeled with two fluorescent dyes, which does not fit into this small area. We observed that internal probes, which bind to the overlapping region of sense-antisense pairs, visualized an insufficient number of RNAs. Moreover, the closer the probes were located to each other, the more they suffered from bleaching effects.

Consequently, we proceeded with tagged constructs and suitable probes in subsequent experiments. The tag was meant to function as the binding platform for probes. However, an intrinsic problem with tagging e.g. the mRNA endogenously, the tag would also be part of its asRNA as it is generated by pervasive transcription and through this simply extend the dsRNA with the tag. Consequently, the tag could not be used anymore, as it was now part of the dsRNA. Therefore, we transcribed the chosen sense RNA (*PHO85*) and its asRNA individually from plasmids. We could show that the sense and asRNAs found each other and formed a dsRNA with two different tags.

However, there were more pitfalls: Selecting the appropriate tag and probes was complicated too, because e.g. structured RNA such as the MS2 loop containing RNA did not allow probe hybridization either. Ultimately, we succeeded by adding a GFP tag for the sense RNA and a repetitive MYC tag for the antisense RNA. As mentioned above, the fluorescent labels in close proximity caused bleaching of the signal, which is why we switched to Alexa labels that show reduced quenching effects. Nevertheless, the probes could not be designed side by side and we had to leave spaces of at least 10 nucleotides to avoid signal quenching.

[REDACTED]

To explain this “journey” of finding the right conditions to localize dsRNAs in smFISH to the reader within the context of this manuscript would exceed the nature format. Therefore, we decided to present only the final functioning instructions to successfully carry out smFISH in yeast with dsRNA.

[REDACTED]

In conclusion, our smFISH study not only addressed the complexities of double-stranded RNA analysis [REDACTED]. The smFISH studies fully support our current findings and let us recognize its significance and currently unrealized options in RNA biology.

Fig. 3h needs quantification. I am not so convinced the NES is functional as is stated in the text based on the example image.

As requested, all microscopy experiments were quantified.

To address the concerns that the NES might not be functional we now present not only the cytoplasmic localization of the tagged NES-tagged RNaseIII, but also a new assay that confirms functionality of the enzyme. We blocked translation to enrich dsRNAs in the cytoplasm, which was again reduced when RNaseIII-NES was expressed (see new Extended Fig. 7n).

Regarding Fig. 4a, if I understand correctly the IP is dsRNA-dependent. Is the interaction indeed lost when the sample is treated with RNase?

To analyze that, we repeated the J2-IP with and without the addition of RNaseIII and indeed, found a loss of Dbp2 interaction when RNaseIII was added (see new Fig. 4e). We thank this reviewer for the excellent suggestion!

Again, quantification of microscopy is appropriate for Fig. 4b.

As requested, we added quantifications to all microscopy experiments (see new Fig. 1n, 2c, 3d, 4c, 4g and Extended Figure 7b).

Referee#2:

The paper by Coban et al. presents interesting novel evidence that the formation of double-stranded RNA (dsRNA) by hybridization of mRNA with an antisense RNA may increase mRNA export efficiency.

In Fig. 1, the authors use nucleo-cytoplasmic fractionation to show that mRNAs described to form dsRNAs accumulate more efficiently in the cytoplasm, and their export is affected by *mex67-5* and *xpo1-1* export receptor mutations. Interestingly, over-expression of a *PHO85* antisense RNA (asRNA) from a plasmid promotes the *PHO85* mRNA cytoplasmic localization.

In Fig. 2, the authors use immunofluorescence with the J2 antibody specific for dsRNA to show that cytoplasmic dsRNA increases in *nmd3Δ* and *rpl10* mutants, suggesting that cytoplasmic dsRNA levels are modulated at the translational level. Another experiment uses RIP to examine the binding of ssRNA or dsRNA to ribosomal protein Rps2 after release of the temperature sensitive *mex67-5 xpo-1* mRNA export mutant from 37 °C to 25 °C. The data indicate that dsRNAs become bound to Rps2 faster than ssRNAs, suggesting preferential export or recognition by ribosomes. Moreover, well-designed EMSA competition experiments provide evidence that the purified Mex67-Mtr2 heterodimeric export receptor preferentially binds dsRNA.

Fig. 3 examines dsRNA formation in cells submitted to stress. Using J2 immunofluorescence, they show cytoplasmic accumulation of dsRNA in *set2Δ*, a mutant promoting asRNA transcription. Nucleo-cytoplasmic fractionations combined with RNA-seq experiments further indicate that *set2Δ* promotes asRNA production and stimulates export of sense mRNAs.

Similar phenotypes are observed following salt or ethanol stress. In Fig. 3, the authors manipulate the localization of RNase III and provide further evidence that stress leads to formation and export of dsRNA in the cytoplasm that is important for cell survival.

Finally, Fig. 4 provides strong evidence that loss of the helicase Dbp2 interferes with dsRNA formation and the enhanced export of PHO85 mRNA upon PHO85 asRNA induction.

This paper uncovers very interesting new ideas and potential mechanisms; however, some experiments are poorly described and not always well controlled. The conclusions drawn from the observations are sometimes too strong. For this work to be considered for publication in Nature, additional experiments are required to validate the proposed model.

We thank this reviewer for the general consideration of our study in Nature and we have added multiple experiments that support our conclusions.

Major comments:

1. Many conclusions are based on immunofluorescence experiments using the J2 antibody described to recognize dsRNA of >40 bp. However, no control experiment in the paper really shows what type of dsRNAs are recognized by this antibody; does the signal mostly correspond to mRNA-asRNA hybrids or to other types of dsRNAs?

The J2-antibody was already used in *E.coli* to identify its dsRNA transcriptome (Lybecker et al., 2014). In this organism, only sense antisense pairs were identified, but not for example

rRNA that has double stranded regions. Obviously, these regions are all below the length needed for detection with the J2 antibody.

We assumed that this would be the same for yeast, however, did not show this. Therefore, we carried out J2-RIPs and subsequent RNA-seq analyses in yeast to identify the bound RNAs. This generated an additional dataset for dsRNAs in yeast, which we incorporated into our analyses along with the RNAi-seq data (see new Fig. 1e-h and Extended Fig. 2). We found that the J2-RNA-seq primarily enriches asRNAs, always alongside with its sense transcript, which indicates that it specifically binds to dsRNA. Clearly, not every mRNA showed an equal enrichment, as sense transcripts are mostly present with higher amounts. If an asRNA constitutes only a fraction (e.g., 1/10) of the corresponding mRNA quantity, the asRNA may be enriched, while the majority of the mRNA remains single-stranded and is not J2-bound.

Reassuringly, the J2-RNAseq and the RNAi-seq data sets are highly similar which shows that we indeed identified the dsRNA transcriptome. The targets in the J2-RNAseq and thus the double-stranded targets are also enriched in the cytoplasm, which reinforces the statements we have already made (Fig .1h). Importantly, like seen in *E.coli* also in yeast the J2-RIP did not enrich rRNA, which supports the assumption that only longer RNA double strands as in sense-antisense pairs are recognized.

There is no mention of how many mRNAs may form dsRNAs due to antisense (as) transcription under normal or stress conditions.

This is an interesting comment and we would like to explain why we did not discuss numbers in our manuscript initially.

Those RNAs that were detected to be dsRNAs in both, the J2-seq and the RNAi-seq come from sense-antisense pairs. Mostly, this method enriched asRNAs as they are normally the minority within the ds-pairs. Therefore, it is not possible to determine dsRNA through their

enrichment as we cannot identify those transcripts that form a double strand to a small extent but are mainly present in the single strand.

Another approach to identify the potential amount of dsRNAs would be to assume that all pairs are in a double strand; simply by counting numbers of both transcripts. For this, the single cell analyses of Ribelles and colleagues were particularly helpful and were now included in the new Fig. Extended 3a in the manuscript. Their RNA-seq data set shows that the potential status of an mRNA as dsRNA or ssRNA differs in different cells. And this makes sense because cells adapt differently to different conditions and this is very individual. When the expression of a gene is induced, for example in response to stress or carbon source switching, the first mRNAs are preferentially exported as dsRNA to reach their desired expression level faster, but under steady expression, less boosting will occur and the advantage of the dsRNA in this function falls back. In this situation most mRNAs are likely present as ssRNA. From their data set it became only clearer that a) cells react individually and b) that asRNAs are increased to boost gene expression.

Another layer of information is important in this context. Upon nuclear export of dsRNAs the pairs are separated by translation. The asRNA is degraded, while the mRNA is still present in cells as a ssRNA, although it was a dsRNA before. Therefore, we are reluctant to state how many RNAs (have) form(ed) a dsRNA. Essentially, every mRNA is able to be in a dsRNA at a certain condition, cell state and a defined time.

More generally, how long should the mRNA-asRNA hybrids be to be efficiently exported?

This is also an intriguing question on which we can only speculate at the moment. What we can certainly add is the average length of the overlap of the sense and asRNA in yeast by their annotation (see new Extended Fig. 3f).

[REDACTED]

To be able to give clear statements of how the design for a boosting or an inhibitory dsRNAs has to be, we certainly have to carry out further extensive investigations and systematic studies. Nevertheless, we found that the minimal naturally overlap in yeast is 240 bp, the average overlap is around 800 bp and the longest overlap is approximately 2400 bp (see new Extended Fig. 3f)

2. There should be additional experiments showing the presence of both sense mRNAs and antisense transcripts in the same cells by single-molecule FISH to get an idea of the prevalence of this mechanism under normal versus stress conditions.

a) More specifically, Fig. 1 should include smFISH experiments of PHO85 sense and antisense RNAs in the presence of p empty or p PGAL1:SUT412.

b) Fig. 3 should also include smFISH experiments revealing specific sense and antisense RNAs (i.e. SEG2 and others) in wt, set2 Δ , and specific stress conditions. c) Finally, Fig. 4 should include smFISH experiments of PHO85 sense and antisense RNAs in wt and dpb2 Δ in the presence of empty plasmid or PGAL1:SUT412.

As already pointed out for reviewer 1, we thought that this was a very good idea and we started experiments right away, but as we learned over the time the smFISH with dsRNAs was a highly challenging experiment that took us quite some time to establish, yet yielded in numerous important insights. Generally, for successful usage of this technique, probes must hybridize with the RNA. However, when complementary RNA strands (sense and antisense) are already hybridized to each other, the probe does not bind. As already identified by Wery and colleagues (2016) through the RNAi-seq data and confirmed by our J2-seq analysis,

double stranded RNAs that are formed in yeast are predominantly formed over the entire length of both, the sense and asRNA. This leaves only minimal overhangs on the pairs for possible probe annealing, as these overhangs are usually not longer than ~100 nucleotides. To visualize a single RNA in yeast a set of at least 12 probes are required, each labeled with two fluorescent dyes, which does not fit into this small area. We observed that internal probes, which bind to the overlapping region of sense-antisense pairs, visualized an insufficient number of RNAs. Moreover, the closer the probes were located to each other, the more they suffered from bleaching effects.

Consequently, we proceeded with tagged constructs and suitable probes in subsequent experiments. The tag was meant to function as the binding platform for probes. However, an intrinsic problem with tagging e.g. the mRNA endogenously, the tag would also be part of its asRNA as it is generated by pervasive transcription and through this simply extend the dsRNA with the tag. Consequently, the tag could not be used anymore, as it was now part of the dsRNA. Therefore, we transcribed the chosen sense RNA (*PHO85*) and its asRNA individually from plasmids. We could show that the sense and asRNAs found each other and formed a dsRNA with two different tags.

However, there were more pitfalls: Selecting the appropriate tag and probes was complicated too, because e.g. structured RNA such as the MS2 loop containing RNA did not allow probe hybridization either. Ultimately, we succeeded by adding a GFP tag for the sense RNA and a repetitive MYC tag for the antisense RNA. As mentioned above, the fluorescent labels in close proximity caused bleaching of the signal, which is why we switched to Alexa labels that show reduced quenching effects. Nevertheless, the probes could not be designed side by side and we had to leave spaces of at least 10 nucleotides to avoid signal quenching.

[REDACTED]

To explain this “journey” of finding the right conditions to localize dsRNAs in smFISH to the reader within the context of this manuscript would exceed the nature format. Therefore, we decided to present only the final functioning instructions to successfully carry out smFISH in yeast with dsRNA.

[REDACTED]

In conclusion, our smFISH study not only addressed the complexities of double-stranded RNA analysis but also provided deeper insights into antisense-mediated preferential export. The smFISH studies fully support our current findings and let us recognize its significance and currently unrealized options in RNA biology.

In the case of the stress-induced sense antisense pairs, we could not use the galactose inducible promoter, because with it there would be no stress response. Therefore, and to be as close to the natural context as possible, we tagged both the asRNA and senseRNA genomically. **[REDACTED]**. As much more research is needed to explain how pervasive transcription and in particular the synthesis of asRNA works, we have no solution for this. Also, we did not find another stress responsive gene that has naturally long overhangs on the sense and antisense pairs that we could place enough probes to visualize the single RNAs in cells. Therefore, we have to rely on the detection of bulk RNAs with the J2 antibody and the high throughput analyses under these conditions as presented in Fig. 3.

It would also be very interesting to perform the same experiment following Dbp2 over-expression. If the model is correct, one should be able to observe an even greater increase in PHO85 mRNA export compared to wt when PHO85 asRNA is induced.

[REDACTED]

As we are still trying to unreveal the underlying regulatory circuit, we would not like to include the experiment in the manuscript, to avoid confusion, but can certainly do this upon request.

Minor corrections and questions to be addressed:

Line 58: these cytoplasmic lncRNAs

No longer applicable

Line 63: corresponding (regarding) mRNAs

No longer applicable

Fig. 1c: How many different sense and antisense transcripts are considered in this analysis; is the C/N ratio based on the same sequencing as in Fig. 1b?

The C/N ratio in Fig. 1d (former 1c) is based on the same RNA-seq as displayed in Figure 1a and b. We added absolute numbers and declared the usage of the displayed data for better understanding.

Based on these sequencing data, how should one evaluate the number of cells producing just sense, AS or both?

mRNA amounts clearly exceed asRNA amounts for most genes. Therefore, an mRNA is most likely found to be single stranded. In single cell analysis about 10 % of the mRNAs are simultaneously present in the same cell with their asRNA. Conversely, 50% of the asRNAs

are present in the same cell as its senseRNA. This fits well into our model, as the preferential export (boosting) should only occur for specific set of genes that need to be upregulated/adjusted at a certain time and state. We have made this clearer now in our model and added the following text to the manuscript:

86 “The predominant enrichment of asRNAs is in line with the observation that asRNAs are on average expressed at 10-fold lower levels compared to the coding transcripts¹⁷ and simply have a higher probability of being present in a dsRNA. Moreover, analysis in single cells reveal that over 90% of the sense RNAs are not at the same time present as their asRNA but ~50% of the asRNAs are present with their sense transcript (Extended Fig. 3a).”

Line 78: How many AS RNAs are more abundant than the corresponding mRNA, which is more cytoplasmic than an mRNA having lower levels of AS.

We added total numbers to both groups (see new Fig. 1d).

How many different sense-AS pairs are these analyses based on. How were these analyses done?

Sense-antisense pairs were identified with BEDtools intersect. The expression level of the sense and antisense transcript was then determined based on the RPKM from our Cytoplasmic fractionation RNA-seq and compared with each other.

518 “Overlapping features respectively sense and antisense pairs were identified with BEDTools intersect⁵³ requiring overlaps to occur on the opposite strand with a minimum overlap of 0.5. lncRNAs were only considered in analysis as SUTs, XUTs or CUTs if they do not overlap with other transcripts of the other types on the same strand.”

Fig. 1d: What sequencing data were used? How many dsRNAs (s/as pairs) were in each of the 10 groups? This information is not mentioned in the Materials and Methods.

The transcripts were grouped based on the RNAi-seq of Wery et al., 2016. Then the mean value from the enrichment in our Cytoplasmic-fractionation RNA-seq of each group was calculated and displayed. We added the total numbers of transcripts to each group (see new Fig. 1d).

Fig. 1d, “the more an RNA is prone to form a double strand the more it is localized to the cytoplasm in wild type cells”: Is this true for both mRNAs and asRNAs?

The groups shown in the new Fig. 1i include lncRNAs and mRNAs. Thus, it is true for both groups of transcripts. mRNAs alone also show this trend. In asRNA transcripts, there are differences between CUTs, XUTs and SUTs, which we address in a new figure. Here it can be seen that XUTs in particular are cytoplasmically enriched. (see new Extended Fig. 3g).

Fig. 1e, “all groups of RNAs, including the dsRNAs were shifted to the nucleus”: Are all RNAs affected the same? Based on the histogram, it seems that the RNAs with the highest probability to form dsRNA are less affected by *mex67-5 xpo1-1*; is that right?

Yes, that is right. There seems to be a small group of dsRNAs, that have an additional cytoplasmic stabilization, which becomes visible after blocking the export. We assume that these RNAs are not translated immediately after export, which is typical for RNAs that have to reach a cellular destination before translation for example. A GO-term analysis indeed showed, that these RNAs code for proteins that are expressed at the cell periphery and membrane. Due to elongated cytoplasmic transport, these RNAs stay longer in the cytoplasm and therefore behave differently than other mRNAs.

Still, it is interesting, that these kinds of mRNAs also tend to be more in double strands than others. But thinking about the encoded proteins, cell membrane proteins are usually also involved in signal transduction, the uptake of nutrients and cell signaling and thus might have been boosted but did not reach their destination yet. We omitted to discuss this in the manuscript due to the space limitations, but we can certainly add such a discussion upon request.

Fig. 1g: y-axis should say PHO85 mRNA instead of PHO5 mRNA (same in Fig. 4d).

We thank this reviewer for uncovering this mistake and we corrected it accordingly (see Fig. 4h).

There is the same amount of unspliced in total cell and cytoplasm when overexpressing PHO85 asRNA, which is unexpected. One would expect more unspliced in the total vs cytoplasmic fraction, since unspliced should be retained in the nucleus. How does it look in the presence of p empty, when PHO85 asRNA is not over-expressed?

In this figure, we analyse the change in unspliced *PHO85* mRNA in both the total and cytoplasmic fractions after overexpression of asRNA. The similar reduction in both fractions does not mean that the same amounts of unspliced RNA are present, but only that the change is very similar. After *asPHO85* expression we see a slight decrease in the unspliced *PHO85* mRNA amounts. The change is not significant, but at least splicing does not seem to be affected.

Ext. Data Fig. 2: The quantification of dsRNA vs ssRNA strand-specific cDNA synthesis and qPCR has to be explained in more detail.

We explained the method in more detail.

“To exclusively measure either mRNA or asRNA in qPCR, RNA specific reverse primers were used (Table 5) in cDNA synthesis (Nippon genetics) and two separate samples were created. Each of them contained either the asRNA primer or mRNA primer, which resulted in separate asRNA and mRNA cDNAs. Furthermore, actinomycin D was added together with the reverse transcriptase as it prevents unspecific transcription from DNA and thereby secures strand specific transcription as reported previously 46.47. In the qPCRs the corresponding cDNA of mRNA and asRNA from one gene were measured with the same primer pair.”

Fig. 2a: As already raised above, what is known about the nature and length of dsRNAs revealed with the J2 antibody? What fraction represents mRNAs compared to other types of transcripts?

Initially, we analyzed the length of the formed dsRNAs of mRNAs and asRNA based on Wery et al. 2016 (Extended Fig. 3c, e). Additionally, we carried out J2-RIP experiments followed by RNA-seq to identify the bound dsRNAs directly (see new Fig. 1e-g and Extended Fig. 2). Reassuringly, RNAi-seq and J2-RIP-seq were highly correlating with a spearman correlation of 0.72 (see new Extended Fig. 3b, 5c). As asRNAs are less expressed than their senseRNA, naturally more asRNAs were identified to be elevated, but we could also find mRNA to be enriched in the J2-eluate.

Neither in the J2-RIP-seq in *E. coli* (Lybecker et al., 2014), nor in our J2-RIP-seq carried out in yeast, other RNA species were enriched.

The dsRNAs seem to accumulate within cytoplasmic foci, which become more frequent and intense in the *nmd3Δ* and *rpl10* mutant backgrounds (and even more so in *set2Δ*, Fig. 3a). What do these aggregates/condensates correspond to?

The speckle-like staining of the J2 antibody in the cytoplasm is most likely due to dsRNAs that await translation. Such “J2”-bodies were to our knowledge not described before. We investigated whether they are overlapping with P-bodies or stress granules by co-localization of the regarding marker proteins Edc1 and Pub1, respectively, but could not detect any overlap. Due to the space limitations, we did not add this negative result to the manuscript, but can certainly do this upon request.

As the J2 antibody recognizes dsRNA sequences of around 40 nt length, several antibodies bind to one dsRNA. On a dsRNA of for example 1200 bp, 30 antibodies can bind. The Cy3-coupled secondary antibody will thus concentrate the signal on this restricted area which might lead to the speckle like staining.

Why is there also more polyA cytoplasmic signal in *nmd3Δ* and *rpl10* mutants? Are the dsRNAs more stable? Are the dsRNAs adenylated?

In the mutants *nmd3Δ* and *rpl10(G161D)* translation is impaired. This effects ssRNA and dsRNA at the same time. Both accumulate within the cytoplasm and have a poly(A) tail that is detected with an oligo d(T) probe as well. Also lncRNAs were described to have a poly(A)tail (van Dijk et al., 2011)

Fig. 2b: Based on what criteria have the ss- and dsRNAs analyzed in this experiment been chosen?

dsRNAs were chosen by three criteria:

1. They are enriched in RNAi-seq (Wery et al., 2016)
2. The asRNA is higher expressed than the mRNA, based on RNAseq
3. The mRNA co-precipitates with the J2 antibody

ssRNAs were chosen by the opposite criteria

1. They are not enriched in RNAi-seq
2. They have no expressed asRNA

We added the following text to the material and methods section:

473 “For qPCR measurements the single stranded RNAs *RPS17A*, *RPS6A* and *TDH1* and the dsRNA *FRE5*, *HPF1* and *PRY3* were analyzed. dsRNA targets were chosen based on three criteria: The asRNA had a higher RPKM than the sense RNA, they were identified as dsRNA in RNAi-seq experiment and enriched after J2 pulldown (Extended Data Fig. 2a).”

Why are there seemingly two waves of dsRNA binding to Rps2 following the shift from 37 °C to 25 °C? It increases until 10 min followed by a decrease, but it goes up again at 60 min.

The export of RNAs is mediated by Mex67 that is recruited by multiple RNA-binding proteins (RBPs) including the guard proteins, upon quality control. They are either released from the RNA directly after export or during translation. Subsequently, they are reimported into the nucleus for another round of transport. This circuit maintains a sensitive balance between the cytoplasmic and nuclear pools of the RBPs and Mex67. We hypothesize that when export is blocked and then released, an unusual large quantity of RBPs is exported simultaneously, temporarily disrupting the equilibrium between the nuclear and cytoplasmic RBP content. After the initial export round, the cytoplasmic RBPs must undergo recycling and re-import, a process that takes time and leaves newly transcribed nuclear RNAs unattended. This might result in the decline observed at 30 minutes. After 60 minutes, the equilibrium between nuclear and cytoplasmic RBP-content is re-established and returns to its initial state.

The standard deviations of these measurements are quite big depending on the time point.

Why is that?

The boxplots of ssRNAs and dsRNAs comprise three different targets, which show the same tendency but in different intensities. This leads to relatively high standard deviations. As this reflects their natural situation, we left the figure unaltered.

Fig. 2c: The EMSA experiment is not well described, neither in the text nor in Materials and Methods. What length are the ss- and dsRNAs used in this experiment? How many Mex67-Mtr2 heterodimer molecules can bind the ss- or dsRNA? What are the absolute molarities used in this experiment?

To improve the description of the EMSA, Oligos used in the EMSA are now in a separated table (Table 6) and we have extended the explanation in the Material and Methods as well as in the main text. These parts now also indicate the length of the oligos used and the amount of Mex67-Mtr2 bound.

430 “Every RNA contained 36 nucleotides and had the equal amount of C, G, T and A (Table 7).”

191 “A 2-molar excess of Mex67-Mtr2 fully upshifted ssRNA, and was saturated at 5-molar excess (Fig. 2g). In contrast, dsRNA reached binding saturation at a 10-molar excess, with a full upshift at 3-molar excess. This indicates that dsRNAs can bind about two heterodimers every 7 bp, while ssRNAs accommodate only one within the same length.”

The molarities were calculated as follows: The molar excess of Mex67-Mtr2, that led to the saturation of the transcripts, represents their suggested binding capacity. For instance: dsRNA showed its maximum upshift at 10-molar excess = the binding capacity of the 36mer dsRNA is around 10 molecules of the heterodimer Mex67-Mtr2.

It would also be useful to put a size marker to indicate how the ssRNA and dsRNA EMSA assays compare.

The EMSA is done in a non-denaturing gel, thus, molecules and complexes do not run corresponding to their size but also their charge. We think a size marker wouldn't be appropriate. But we added an additional experiment with switched fluorescent dyes to support our results (Extended Fig. 6g) with the same result. dsRNA shows a preferential and a higher binding of Mex67-Mtr2.

Fig. 2d: p. 8 line 189, It should say Fig. 2d and not Fig. 2e (there is no Fig. 2e).

We thank this reviewer for the critical reading.

Fig. 3e: p. 9 line 214 “the mean of the sense RNAs rather stayed the same and showed specific sense transcript upregulation”: How many sense and asRNAs were considered in this analysis? How many genes show a substantial increase in antisense transcription upon shift to 0.6 M NaCl?

Based on the RNA-seq data of Ho et al., there are 43 significantly increased mRNAs, which had also a significantly changed asRNA expression. 42 had a significantly asRNA transcription and 2 had a decreased asRNA (see new Fig. 3b).

Further Go-term analysis of all significantly increased asRNAs during osmotic stress show, that their corresponding genes are involved in response to stress and other stimuli. Due to the space limitations, we did not include this description in the text, but can certainly do so upon request. Ribelles et al., 2014 found 41 out of 91 sense-antisense pairs to be positive correlated under osmotic stress induction.

Which specific sense transcripts were upregulated, and was their upregulation also enhanced in *set2Δ*?

The transcriptome in *set2Δ* was analyzed by Venkatesh et al., 2016. There, mostly antisense transcription is enhanced and not sense transcripts. asRNA transcripts increased in *set2Δ* are antisense to genes involved in osmoregulatory sensing and aging, which was found by Venkatesh and colleagues.

Fig. 3h: The dsRNA signal in the wild type appears much stronger than in other wild-type controls such as Fig. 2a, 3a, or 3f.

In this experiment, we slightly increased the post-processing to emphasise the absence or reduction of dsRNA in cells overexpressing RNaseIII constructs. But we agree with this reviewer that this could be confusing to the reader. Therefore, we chose now to present an uniform procedure for signal admission.

Fig. 4a: Describe in the legend whether these cells were grown in galactose to induce MYC-DBP2 before J2 CoIP.

We switched the promoter to the endogenous DBP2 promoter. We think, the experiment is better this way.

p. 12: There is a wording problem: "Clearly, in the absence of Dbp2 the previous effect that overexpression of the asRNA shifted the localization of the sense RNA to the cytoplasm (Fig. 4c, 4d)".... was abrogated?

We rephrased the sentence:

"Clearly, although the asRNA was still highly enriched in *dbp2Δ*, it was not able to manipulate the localization of its sense RNA (Fig. 4f-h, Extended Data Fig 8f, 8g), suggesting that Dbp2 is the key factor for dsRNA formation that enables preferential export of dsRNAs."

Referee#3:

Krebber and co-workers describe a novel and intriguing role for asRNA in yeast: facilitation of nuclear export of their sense mRNA counterparts. The authors perform transcriptomic and traditional molecular biological experiments and find that dsRNA formation between asRNA and their complementary mRNA counterparts promotes export of these mRNAs. They further implicate the RNA helicase Dbp2 in the formation of the dsRNA.

The manuscript reports a foundational discovery with potentially far-reaching implications for our understanding of how gene expression is regulated at the RNA level. The data also provide a novel, largely unanticipated explanation for the role of asRNAs, a longstanding question in biology. The experiments are well rationalized, and approaches and data are well explained.

Strengths of the study include several well-conceived and executed orthogonal approaches demonstrating the claimed link between asRNA-mRNA binding and promotion of mRNA export. Weaknesses of the study are potential problems in the assignment of a role for Dbp2 in the formation of the dsRNA, and the heavy reliance on the J2 antibody for the detection of dsRNA in cells, as outlined in more detail below. It seems advisable to address these and some more minor issues (also outlined below) prior to publication. Assuming that the weaknesses are addressed, the manuscript is likely to appeal to the broad readership of Nature.

We thank this reviewer for acknowledging our findings and helping us improve our work.

Comments / major issues:

1) The final major point of the manuscript is the suggestion that the RNA helicase Dbp2 promotes formation of the dsRNA between the mRNAs and their corresponding asRNAs. Based on the data presented, the claim that Dbp2 plays a direct role in the dsRNA formation appears premature, for the following reasons:

a) The first key experiment is a co-IP of Dbp2p with the dsRNA binding J2 antibody (Fig. 4a), based upon which the authors claim evidence for binding of Dbp2 to dsRNA in vivo (lines 261-262). This claim is an overly optimistic interpretation of the data. Dbp2 can bind to any RNA (structured or unstructured) associated with a dsRNA region, to other proteins associated with any such RNA region, or a combination thereof. Given that alternative scenarios are consistent with the observed data, direct evidence is needed to conclusively demonstrate that Dbp2 binds to dsRNA in vivo. Since Dbp2 is a DEAD-box helicase that at least transiently binds dsRNA in order to unwind duplexes, some level of dsRNA binding is inherently expected. The point here is that the data shown are insufficient to support dsRNA binding by Dbp2 in vivo.

To address this point, we repeated the J2-IP experiment but added recombinant RNaseIII, which degrades dsRNA and lost co-precipitation of Dbp2 (see new Fig. 4a). The J2 antibody recognizes dsRNAs of a length of at least 40 bp. Since the telomerase RNA *TLC1* is the only known transcript to form an intramolecular dsRNA of 40 bp, there are no other targets that should be recognized by J2. Only dsRNAs formed by mRNA and asRNA hybridization exceed the length for J2 recognition. This was also confirmed by genome wide J2-RIP-seq experiments. (Fig. 1e-g, Extended Fig.2)

b) The authors then use a *dbp2* deletion strain to examine changes in the accumulation of dsRNA in the cytoplasm. Here again, the data are consistent with the claim that Dbp2 is a key

factor for ds formation and preferential export (lines 272-273). A direct role of Dbp2 in dsRNA formation is clearly implied by the authors, but Dbp2 might work well upstream of the actual dsRNA formation. Dbp2 is well known to act on a multitude of RNAs, and its deletion likely affects many processes, which might include production of one or more factors directly involved in dsRNA formation. More specific evidence for a direct role of Dbp2 in dsRNA formation appears necessary, especially given that Dbp2 has been implicated in the unwinding of asRNA-mRNA complexes (PMID: 26805575), and because of the pleiotropic effects of a *dbp2* deletion.

Wery et al. 2016 (PMID: 26805575) showed the stabilization of two asRNA XUTs upon loss of *DBP2*. Since Dbp2 is a known helicase, they concluded, that this stabilization is due to the remained dsRNA structure with their mRNA. But it could also be interpreted differently. We propose, that the asRNA is stabilized as a single stranded RNA due to the prevented dsRNA formation and subsequent nuclear export. By preventing dsRNA formation in *dbp2Δ* the normal export and degradation of these XUTs in the cytoplasm is inhibited, thereby this transcript accumulates in the nucleus. This could be the alternative explanation with the assumed defect in the dsRNA formation activity in *dbp2Δ* and would also explain why the amount of two XUTs is increased while the asRNA SUT decreased in the experiment of Wery et al..

To further support the dsRNA formation activity of Dbp2, we added five different additional experiments that show that Dbp2 is a dsRNA forming helicase in yeast. First, we show that in *dbp2Δ* dsRNA amounts are strongly decreased as judged by J2 immunofluorescence and secondly by dot blot (see new Fig. 4a, c, d). Thirdly, we show in smFISH that the cytoplasmic co-localization of the *PHO85* sense and asRNA is lost in *dbp2Δ* (see new Fig. 4f, g) and fourthly that RNAs that were increasingly identified to be part of a double strand become more sensitive to DMS labelling in *dbp2Δ* compared to wild type (see new Extended Fig. 8e). Lastly,

we also added experiments with the helicase Mtr4, which was identified to regulate XUT expression when mutated, very similar to what was suggested for Dbp2 in the recent Wery et al. paper (Wery et al. 2023, Front RNA Res). We show that the helicases show different phenotypes and identify Mtr4 as a dsRNA degrading helicase and Dbp2 as the dsRNA forming helicase (see new Fig. 4a-c).

Together, all experiments support the identification of Dbp2 as the dsRNA forming helicase. Moreover, we have evidence for a role of its co-factor Yra1 in the same process. However, because they regulate each other, things are more complicated and we avoided to include these rather preliminary data in this manuscript, because a) this would not fit into the short format and b) needs further investigations, as Dbp2 and Yra1 regulate each other via their introns, which is beyond the scope of this paper. With our new data we think that we have strengthened our evidence for a role of Dbp2 in dsRNA formation *in vivo*.

The use of glucose deprivation, which moves Dbp2 to the cytoplasm, might alleviate some confounding issues. In addition, there are several CLIP datasets for Dbp2-RNA binding in yeast. Examination of whether and how the Dbp2 binding sites in cells correspond to mRNA-asRNA binding regions might be instructive for assigning a physical function of the protein in dsRNA formation.

In contrast to the published cytoplasmic shift of Dbp2, we did not see a cytoplasmic localization of Dbp2 upon glucose starvation (see below).

We left this figure out due to the space limitations, but can certainly include this figure into the manuscript upon request.

To analyze the published CLIP data set was an excellent suggestion and we analyzed the iCLIP data of Dbp2 and structure seq analysis in *dbp2Δ*. The iCLIP data show binding of Dbp2 to all RNAs, whether or not they are potentially double stranded or single stranded (see new Extended Fig. 4b), suggesting that Dbp2 initially contacts all transcripts, possibly for unwinding as suggested earlier (Ma et al., 2014). We assume that the formation of a dsRNA depends a) on the presence of the regarding asRNA and b) the co-factor of Dbp2 Yra1.

2) The authors rely heavily on the J2 antibody to detect dsRNA in cells. The relevant microscopy data are very qualitative, and it is not immediately apparent how robust the microscopy data are. In some cases, the microscopy experiments are validated by specific, gene-focused experiments. In other cases, the microscopy data stand alone. As the localization of dsRNA is a central point in the ms, the microscopy data should be quantified, and robustly validated in all cases by an orthogonal approach.

To validate the specificity of the J2 antibody, we sequenced the RNA of the eluate and the unbound supernatant in a J2-RIP-seq. (Fig. 1e-g and Extended Fig. 2) To demonstrate the robustness of the microscopy, we quantified cells of three replications in the different experiments and show them in an additional figure. Furthermore, we support the conclusions drawn from the microscopy with the J2 antibody by further experiments and hope that they are now more convincing. (see new Fig. 2d and 4d).

Comments/minor issues (no particular order):

3) Line 94: The authors indicate that mRNAs with long half-lives accumulate in the nucleus, implying that these mRNAs are not preferentially translated. This observation seems to contradict observations by other groups showing increases in half-lives with preferential translation of mRNAs (e.g. PMID: 25768907). Clarification on this issue is needed.

We agree that this issue needs to be clarified. Essentially, an RNA can have multiple properties, that influence its stability. One reason for a long half-life can be a delay in nuclear export and another an efficient long-lasting translation or signal sequences that increase its degradation. Thus, nuclear and cytoplasmic processes influence the half-life of an RNA and can thus lead to a shift of the transcripts into one or the other compartment. We don't think there's any controversy here, just different aspects. If two points of view correlate with each other, it does not mean that one of them can't also correlate with a third. This might also be individual for different genes.

Our finding (Extended Fig. 5b) shows an increased stability of nuclear mRNAs. In fact, a similar model was suggested earlier in which a delayed export of mRNAs was suggested to buffer the burst like transcription of mRNAs, so that a continuously even amount of RNAs reach the cytoplasm for translation (Bahar Halpern et al., 2015). This is particularly true for genes that are frequently transcribed, as many transcripts are produced at once, which are then regulated by the export. We also see this effect in our analysis, as highly expressed

genes in particular are nuclear-enriched. These highly expressed and nuclear transcripts then have a higher half-life on average. By the same logic, we initially identified that mRNAs can be preferentially exported by an asRNA mediated export burst, leading to a cytoplasmic enrichment. Thus, in both cases export determines delayed or boosted gene expression.

4) Issues with data presentation in figures.

a) Fig. 1b: How were the dsRNA and ssRNA quantified? Is the difference in cytoplasmic vs. nuclear distribution statistically significant, and if so, how was this determined?

We now have two ways of differentiating between dsRNAs and ssRNAs. On the one hand based on the RNAi-seq and on the other hand based on the J2-RIP-seq we performed. Using the J2-RIP-seq, we can divide RNAs into significantly enriched and significantly depleted. Whereby significantly enriched corresponds to dsRNAs, as the J2 binds dsRNA, and significantly depleted corresponds to ssRNAs. Using RNAi-seq, transcripts were categorised into groups based on the amount of degradation products produced by the RNAi system, which is related to the amount of dsRNA. This categorization reflects the probability of a transcript, how often it is present in the double strand. Both classifications clearly show that dsRNA targets are more cytoplasmic than ssRNA targets. On justified request, we have now calculated the significance among these groups. The significance among the groups based on RNAi-seq was now calculated using the Kruskal-Wallis test, which shows a high significance.

We would like to add, that it is likely and can be assumed that transcripts are not either present or not present in the double strand in different cells. The mRNAs of genes can be either single-stranded or double-stranded depending on the cell, the condition and the process in which it is located. For example, an mRNA is present in translation as a single strand, although it was previously exported as a dsRNA. So in the bulk RNA-seqs we are looking at a mixture of the possibilities, which is why again we think it would be best to talk

about a probability of the dsRNA.

b) Fig. 1c: Is the difference in cytoplasmic vs. nuclear distribution statistically significant, and if so, how was this determined?

See above

c) Fig. 1d: The y-axis notes probability, which is by definition a value between 0 and 1, but negative values are given. Clarification is needed.

The left y-axis now displays the groups based on RNAi-seq, and on the right, the probability of forming dsRNA is added as additional graphical support.

d) Fig. 1e: Is the change in nuclear accumulation statistically significant, and if so, how was this determined? As above, the y-axis notes probability, which is by definition a value between 0 and 1, but negative values are given. Clarification is needed.

The left y-axis now displays the groups based on RNAi-seq, and on the right, the probability of forming dsRNA is added as additional graphical support.

5) The gel-shift experiments (Fig. 2c,d) are of good visual quality but lack sufficient rigor.

a) The authors should report the reproducibility of the data and The EMSA were repeated three times and showed a good reproducibility. We added n=3 to the figure legend. Moreover, we repeated the experiment additional three times with the dyes changed with the same result. We added an EMSA in which the FAM label was placed on the

ssRNA and the Cy5 label on the dsRNA with the same result (see new Extended Figure 2e and f).

b) try to quantify the affinities (Fig. 2c), in order to allow assessment of differences in binding efficiency, rather than qualitatively describe the gel shifts.

The molarities were now calculated as follows: The molar excess of Mex67-Mtr2, that led to the saturation of the transcripts, represents their suggested binding capacity. For instance: dsRNA showed its maximum upshift at 10-molar excess = the binding capacity of the 36mer dsRNA is probably 10 molecules of the heterodimer Mex67-Mtr2.

We added the following text to the manuscript (line x):

A 2-molar excess of Mex67-Mtr2 fully upshifted ssRNA, and was saturated at 5-molar excess (Fig. 2g). In contrast, dsRNA reached binding saturation at a 10-molar excess, with a full upshift at 3-molar excess. This indicates that dsRNAs can bind about two heterodimers every 7 bp, while ssRNAs accommodate only one within the same length.”

Reviewer Reports on the First Revision:

Referees' comments:

Referee #1 (Remarks to the Author):

The authors have revised the manuscript, demonstrating that antisense-sense RNA formation facilitates faster mRNA transport from the nucleus to the cytoplasm. The authors diligently addressed the suggestions and comments, incorporating suggested experiments. I appreciate the authors for their efforts for conducting the smFISH experiments, which have strengthened the manuscript's conclusions. Additionally, the authors have quantified the results of the microscope experiments and introduced evidence of an additional nuclear protein. Furthermore, the authors have provided further analysis of the fractionation RNA-seq data and performed reverse labeling for the vitro experiments. To reiterate, the work adds a new twist to the longstanding question of what the function of antisense transcripts is.

I have a few small comments.

In Figure 1n, the ratio-over-ratio representation can be deceiving and is possibly less intuitive to follow. It might be worth considering replacing it with Extended Figure 4a, as it depicts the differences better and easier to understand.

The authors introduce new dataset to confirm the analysis using RNAi-seq. The original paper needs to be cited <https://pubmed.ncbi.nlm.nih.gov/19745116/>. Wery et al did not establish RNAi in budding yeast but performed the sequencing on the strain. The RNAi-seq does show a striking correlation with J2 antibody.

In Figure 1m, it was not clear whether "-as" refers to plasmid or without induction. Perhaps labelling could clarify this.

In Extended Figure 4e, "pgal-sut412" appears to be the same as "asPho85." Is this the same plasmid as used in Figures 1b, c 1? Perhaps checking for consistency would be beneficial.

The experiment in Figure 2f, as presented, seems to come out of the blue. Also requires more controls as presented such as of whether the mRNA levels or transcription of sense not affected? Also it states in legends that the experiment is quantified, but no quantification is shown.

In Figure 4a, the nuclear increase in polyA dbp1 mutant is not clear and not quantified, if I read correctly.

For Figures 4f, g, h, did the authors also determine the absolute expression levels of Pho85? There seems to be a mislabel of "as-sense."

The citation for Venkatesh is misspelled.

Referee #4 (Remarks to the Author):

The revised manuscript by Coban et al., which describes an intriguing role for asRNA in increasing export efficiency of mRNAs in *S. cerevisiae* under certain cellular conditions, has significantly improved with the addition of the further data. The authors have added significant data that further supports their model and have adequately addressed all reviewers' concerns in this revised version. I believe that this manuscript will be of significant interest to a broad readership and recommend its publication in Nature.

I would also suggest that the authors edit the manuscript for further clarity before publication, and especially omit qualitative statements in the text – see examples below.

Line 96: please change “earlier” to “Previous work by...” for clarity

Lines 183 and 199: “Clearly” – qualitative statement - omit

Lines 185-187: “This may be due to a different occupancy of the RNAs by export receptors: As (should be ‘as’) suggested earlier, the more export proteins cover an RNA, the faster transport is initiated.”

Please rephrase this sentence for clarity as it is unclear if the ‘earlier’ in the subclause refers to this manuscript or the ones cited.

Line 229: “Impressively” – qualitative statement - omit

Line 266-267: deletion of > deletions of; also mention if these are single deletions of double deletions.

Line 294: “Reassuringly” – qualitative statement – omit

Author Rebuttals to First Revision:

We are thankful for the valuable comments of the editors and the reviewers that we have fully addressed (see below point-by-point response). We have followed and implemented all comments to provide the manuscript and all related data in the desired form. Please see our specific responses (in black) to the editors and reviewer's comments (in green).

We have implemented the reviewers' comments as follows:

Referee #1 (Remarks to the Author):

The authors have revised the manuscript, demonstrating that antisense-sense RNA formation facilitates faster mRNA transport from the nucleus to the cytoplasm. The authors diligently addressed the suggestions and comments, incorporating suggested experiments. I appreciate the authors for their efforts for conducting the smFISH experiments, which have strengthened the manuscript's conclusions. Additionally, the authors have quantified the results of the microscope experiments and introduced evidence of an additional nuclear protein. Furthermore, the authors have provided further analysis of the fractionation RNA-seq data and performed reverse labeling for the vitro experiments. To reiterate, the work adds a new twist to the longstanding question of what the function of antisense transcripts is.

I have a few small comments.

In Figure 1n, the ratio-over-ratio representation can be deceiving and is possibly less intuitive to follow. It might be worth considering replacing it with Extended Figure 4a, as it depicts the differences better and easier to understand.

We thank this reviewer for the good suggestion. We have swapped the two figures so that the previous extended figure 4a now appears in the main figure as figure 1l.

The authors introduce new dataset to confirm the analysis using RNAi-seq. The original paper needs to be cited <https://pubmed.ncbi.nlm.nih.gov/19745116/>. Wery et al did not establish

RNAi in budding yeast but performed the sequencing on the strain. The RNAi-seq does show a striking correlation with J2 antibody.

We thank the reviewer for pointing this out. That is of course correct, and we are now cited the original paper as well. (line 97)

In Figure 1m, it was not clear whether "-as" refers to plasmid or without induction. Perhaps labelling could clarify this.

We are sorry for the confusion. By "-as" we meant that the cells were transformed with an empty plasmid in contrast to the "+as" in which the plasmid contained the asRNA under the *GAL1* promoter. To counteract a possible misunderstanding and to characterize the condition more clearly, we have now chosen the term "+empty".

In Extended Figure 4e, "pgal-sut412" appears to be the same as "asPho85." Is this the same plasmid as used in Figures 1b, c 1? Perhaps checking for consistency would be beneficial.

The reviewer is right. In order to make this work accessible to a wide readership, we have now decided to refer to the asRNA as *asPHO85* and to omit the gene name *SUT412*. However, when it is named for the first time in the main text, we also mention the gene name *SUT412*.

The experiment in Figure 2f, as presented, seems to come out of the blue. Also requires more controls as presented such as of whether the mRNA levels or transcription of sense not affected? Also it states in legends that the experiment is quantified, but no quantification is shown.

We are sorry for being too brief here. We have now introduced the figure more thoroughly and additionally quantified the protein, the mRNA and the asRNA levels after the asRNA induction. Excitingly, we see a decrease in mRNA with an increase in protein level, which supports our hypothesis of the boosted gene expression (Fig. 2f). The following text was added to the manuscript (see lines 187-189)

In Figure 4a, the nuclear increase in poly(A) dbp1 mutant is not clear and not quantified, if I read correctly.

We calculated that 20 % of the cells in *dbp2Δ* show a nuclear accumulation of the poly(A) RNA. Due to the space limitations of the journal format, we did not add this information to the text but can certainly do so upon request.

For Figures 4f, g, h, did the authors also determine the absolute expression levels of Pho85? There seems to be a mislabel of "as-sense."

We quantified the *PHO85* mRNA levels in the respective experiments utilizing the *dbp2Δ* mutant. The endogenous *PHO85* mRNA from experiment in Figure 4h increases significantly 4.68-fold. In the smFISH, and thus under the control of the *GAL1* promoter, the *PHO85* mRNA level reached 89 % in *dbp2Δ* cells compared to Wild type cells.

Due to the space limitations of the journal format, we did not add this information to the text but can certainly do so upon request.

The citation for Venkatesh is misspelled.

Thank you for the careful reading. We have corrected this accordingly.

Referee #4 (Remarks to the Author):

The revised manuscript by Coban et al., which describes an intriguing role for asRNA in increasing export efficiency of mRNAs in *S. cerevisiae* under certain cellular conditions, has significantly improved with the addition of the further data. The authors have added significant data that further supports their model and have adequately addressed all reviewers' concerns in this revised version. I believe that this manuscript will be of significant interest to a broad readership and recommend its publication in Nature.

I would also suggest that the authors edit the manuscript for further clarity before publication, and especially omit qualitative statements in the text – see examples below.

We thank this reviewer for acknowledging our finding and work for this manuscript. We also thank this reviewer for reading our manuscript so carefully. We have changed all the mentioned passages accordingly.

Line 96: please change “earlier” to “Previous work by...” for clarity

Line 96 and following: “In yeast, Wery and colleagues detected dsRNA formation through an RNAi based screen,”

Lines 183 and 199: “Clearly” – qualitative statement - omit

Line 183 and following: “We found that dsRNAs arrived at ribosomes significantly earlier than ssRNAs, providing further experimental evidence for the preferential export and thus translation of dsRNAs.”

Line 199 and following: “Mex67-Mtr2 dissociated from ssRNA and associated with dsRNA in the presence of increasing dsRNA amounts”

Lines 185-187: “This may be due to a different occupancy of the RNAs by export receptors: As (should be ‘as’) suggested earlier, the more export proteins cover an RNA, the faster transport is initiated.”

Please rephrase this sentence for clarity as it is unclear if the ‘earlier’ in the subclause refers to this manuscript or the ones cited.

Line 186 and following: “This may be due to a different occupancy of the RNAs by export receptors. As suggested by Mofrad and colleagues, the more export proteins cover an RNA, the faster transport is initiated”

Line 229: “Impressively” – qualitative statement - omit

Line 229 and following: “Analysis of the RNA-seq data from Ho and colleagues, that were generated after osmotic shock”

Line 266-267: deletion of > deletions of; also mention if these are single deletions of double deletions.

Line 29 and following: “Interestingly, deletions of the two nuclear helicases Dbp2 or Mtr4 were reported to increase XUT-asRNA levels.”

Line 294: “Reassuringly” – qualitative statement – omit

Line 297 and following: “This binding was lost in the presence of recombinant RNaseIII.”